# Resting-State EEG Alpha Rhythms Are Related to CSF Tau Biomarkers in Prodromal Alzheimer’s Disease

**DOI:** 10.3390/ijms26010356

**Published:** 2025-01-03

**Authors:** Claudio Del Percio, Roberta Lizio, Susanna Lopez, Giuseppe Noce, Matteo Carpi, Dharmendra Jakhar, Andrea Soricelli, Marco Salvatore, Görsev Yener, Bahar Güntekin, Federico Massa, Dario Arnaldi, Francesco Famà, Matteo Pardini, Raffaele Ferri, Filippo Carducci, Bartolo Lanuzza, Fabrizio Stocchi, Laura Vacca, Chiara Coletti, Moira Marizzoni, John Paul Taylor, Lutfu Hanoğlu, Nesrin Helvacı Yılmaz, İlayda Kıyı, Yağmur Özbek-İşbitiren, Anita D’Anselmo, Laura Bonanni, Roberta Biundo, Fabrizia D’Antonio, Giuseppe Bruno, Angelo Antonini, Franco Giubilei, Lucia Farotti, Lucilla Parnetti, Giovanni B. Frisoni, Claudio Babiloni

**Affiliations:** 1Department of Physiology and Pharmacology “Vittorio Erspamer”, Sapienza University of Rome, 00185 Rome, Italy; claudio.delpercio@uniroma1.it (C.D.P.); susanna.lopez@uniroma1.it (S.L.); matteo.carpi@uniroma1.it (M.C.); dharmendra.jakhar@uniroma1.it (D.J.); filippo.carducci@uniroma1.it (F.C.); claudio.babiloni@uniroma1.it (C.B.); 2Oasi Research Institute—IRCCS, 94018 Troina, Italy; rferri@oasi.en.it (R.F.); blanuzza@oasi.en.it (B.L.); 3IRCCS Synlab SDN, 80143 Naples, Italy; giuseppe.noce@uniroma1.it (G.N.); andrea.soricelli@uniparthenope.it (A.S.); marcosalvatore2.segreteria@gmail.com (M.S.); 4Department of Medical, Movement and Well-Being Sciences, University of Naples Parthenope, 80133 Naples, Italy; 5Department of Neurology, Faculty of Medicine, Dokuz Eylül University, 35340 İzmir, Turkey; gorsev.yener@gmail.com; 6IBG: International Biomedicine and Genome Center, 35340 Izmir, Turkey; 7Research Institute for Health Sciences and Technologies (SABITA), Istanbul Medipol University, 34810 Istanbul, Turkey; bguntekin@medipol.edu.tr; 8Department of Biophysics, School of Medicine, Istanbul Medipol University, 34810 Istanbul, Turkey; 9Dipartimento di Neuroscienze, Oftalmologia, Genetica, Riabilitazione e Scienze Materno-Infantili (DiNOGMI), Università di Genova, 16132 Genova, Italy; fedemassa88@gmail.com (F.M.); dario.arnaldi@gmail.com (D.A.); francesco.fama@unige.it (F.F.); matteo.pardini@gmail.com (M.P.); 10Clinica Neurologica, IRCCS Ospedale Policlinico San Martino, 16132 Genova, Italy; 11Neurofisiopatologia, IRCCS Ospedale Policlinico San Martino, 16132 Genova, Italy; 12IRCCS San Raffaele, 00163 Rome, Italy; fabrizio.stocchi@sanraffaele.it (F.S.); laura.vacca@sanraffaele.it (L.V.); chiara.coletti@sanraffaele.it (C.C.); 13Department of Neurology, Telematic University San Raffaele, 00166 Rome, Italy; 14Biological Psychiatry Unit, IRCCS Istituto Centro San Giovanni di Dio Fatebenefratelli, 25125 Brescia, Italy; mmarizzoni@fatebenefratelli.eu; 15Translational and Clinical Research Institute, Faculty of Medical Sciences, Newcastle University, Newcastle upon Tyne NE2 4AE, UK; john-paul.taylor@newcastle.ac.uk; 16Department of Neurology, School of Medicine, Istanbul Medipol University, 34810 Istanbul, Turkey; lhanoglu@kure.com.tr; 17Department of Neurology, Medipol University Istanbul Parkinson’s Disease and Movement Disorders Center (PARMER), 34718 Istanbul, Turkey; drnesrin76@gmail.com; 18Health Sciences Institute, Department of Neurosciences, Dokuz Eylül University, 35330 Izmir, Turkey; ilaydaky@gmail.com (İ.K.); ozbekyagmur1@gmail.com (Y.Ö.-İ.); 19Department of Aging Medicine and Sciences, University “G. d’Annunzio” of Chieti-Pescara, 66100 Chieti, Italy; a.danselmo@unich.it (A.D.); l.bonanni@unich.it (L.B.); 20Department of General Psychology, University of Padua, 35128 Padova, Italy; roberta.biundo@unipd.it; 21Parkinson and Movement Disorders Unit, Study Center for Neurodegeneration (CESNE), Center for Rare Neurological Diseases (ERN-RND), Department of Neuroscience, University of Padua, 35121 Padua, Italy; angelo.antonini@unipd.it; 22Department of Human Neurosciences, Sapienza University of Rome, 00185 Rome, Italy; fabrizia.dantonio@uniroma1.it (F.D.); giuseppe.bruno@uniroma1.it (G.B.); 23Department of Neuroscience, Mental Health, and Sensory Organs, Sapienza University of Rome, 00189 Rome, Italy; franco.giubilei@uniroma1.it; 24Centre for Memory Disturbances, Lab of Clinical Neurochemistry, Section of Neurology, University of Perugia, 06123 Perugia, Italy; lucia.farotti@gmail.com; 25Department of Medicine and Surgery, University of Perugia, 05100 Perugia, Italy; lucilla.parnetti@unipg.it; 26Laboratory of Neuroimaging of Aging (LANVIE), University of Geneva, 1205 Geneva, Switzerland; 27Geneva Memory Center, Department of Rehabilitation and Geriatrics, Geneva University Hospitals, 1205 Geneva, Switzerland; 28Hospital San Raffaele Cassino, 03043 Cassino, Italy

**Keywords:** mild cognitive impairment due to Alzheimer’s disease (ADMCI), resting-state electroencephalographic (EEG) rhythms, cerebrospinal fluid (CSF) biomarkers, exact low-resolution brain electromagnetic source tomography (eLORETA)

## Abstract

Patients with mild cognitive impairment due to Alzheimer’s disease (ADMCI) typically show abnormally high delta (<4 Hz) and low alpha (8–12 Hz) rhythms measured from resting-state eyes-closed electroencephalographic (rsEEG) activity. Here, we hypothesized that the abnormalities in rsEEG activity may be greater in ADMCI patients than in those with MCI not due to AD (noADMCI). Furthermore, they may be associated with the diagnostic cerebrospinal fluid (CSF) amyloid–tau biomarkers in ADMCI patients. An international database provided clinical–demographic–rsEEG datasets for cognitively unimpaired older (Healthy; N = 45), ADMCI (N = 70), and noADMCI (N = 45) participants. The rsEEG rhythms spanned individual delta, theta, and alpha frequency bands. The eLORETA freeware estimated cortical rsEEG sources. Posterior rsEEG alpha source activities were reduced in the ADMCI group compared not only to the Healthy group but also to the noADMCI group (*p* < 0.001). Negative associations between the CSF phospho-tau and total tau levels and posterior rsEEG alpha source activities were observed in the ADMCI group (*p* < 0.001), whereas those with CSF amyloid beta 42 levels were marginal. These results suggest that neurophysiological brain neural oscillatory synchronization mechanisms regulating cortical arousal and vigilance through rsEEG alpha rhythms are mainly affected by brain tauopathy in ADMCI patients.

## 1. Introduction

The National Institute on Aging and Alzheimer’s Association has introduced a framework for the neurobiological diagnosis of Alzheimer’s disease (AD) in both research and clinical settings [1,2]. This framework prioritizes the diagnosis of AD through in vivo biomarkers indicative of brain amyloidosis (“A”), tauopathy (“T”), and neurodegeneration (“N”) [1,2]. In this regard, amyloidosis and tauopathy can be detected in vivo through cerebrospinal fluid (CSF) analysis or positron emission tomography (PET), while neurodegeneration is assessed through structural MRI or fluorodeoxyglucose PET (FDG-PET).

Notably, these biomarker-driven characterizations encompass the entire spectrum of cognitive decline in clinical Alzheimer’s disease, ranging from subjective cognitive complaints to overt dementia, with the intermediate stage of mild cognitive impairment (MCI) positioned in between. Clinically, MCI is described as a syndrome characterized by significant cognitive impairment that does not interfere with the activities of daily living [3,4]. The prevalence of MCI in individuals aged 60 years and older varies widely, from 6.7% to 25.2% [4,5], and an MCI diagnosis is a risk factor for progression to dementia, particularly due to Alzheimer’s disease [6]. According to the joint guidelines of the National Institute on Aging and the Alzheimer’s Association (NIA-AA) [7], a clinical diagnosis of MCI requires evidence of a cognitive decline reported by the patient, an informant, or a clinician; objective impairment in one or more cognitive domains; and the preservation of independence in functional abilities. Subtypes of MCI have also been identified, with a focus on memory impairment, the most frequently affected cognitive domain. This has led to a taxonomy of four subtypes, amnesic MCI and non-amnesic MCI, each further classified as single-domain or multi-domain based on the number of cognitive domains affected [8]. Given its high prevalence and the risk of progression to various neurodegenerative conditions [8], the identification of reliable biomarkers for MCI remains a critical area of clinical research.

Furthermore, the current framework model of AD does not address how AD-related neuropathology and neurodegeneration influence the oscillatory neurophysiological thalamocortical mechanisms that regulate cortical arousal and quiet vigilance, which are often disrupted in AD [9,10]. These mechanisms facilitate the integration of postsynaptic potentials in cortical pyramidal neurons, which generate detectable alterations in the electromagnetic fields measured at the scalp level during wakefulness [11]. The measurement of these fields provides insight into ongoing scalp-recorded electroencephalographic (EEG) rhythms during resting-state eyes-closed conditions. Compared to cognitively unimpaired older (Healthy) individuals, AD patients with amnesic mild cognitive impairment (ADMCI) and dementia (ADD) exhibit abnormality in resting-state eyes-closed EEG (rsEEG) rhythms at delta (<4 Hz), theta (4–7 Hz) and alpha (8–13 Hz) frequency ranges [12,13,14].

Several investigations carried out by our international PDWAVES Consortium (www.pdwaves.eu; last accessed on October 2024) have successfully demonstrated that alterations in rsEEG rhythms are associated with various neuropsychological, molecular, neuroanatomical, and pathophysiological markers in ADD and ADMCI patients [13,14].

An unresolved question pertains to the relationship between the rsEEG measures used by our Consortium and brain amyloidosis and tauopathy in AD patients. Such a relationship is expected based on recent studies showing a relationship between biomarkers of brain amyloidosis and tauopathy and rsEEG biomarkers in ADD and ADMCI patients [15,16,17,18,19,20,21,22]. Noteworthy findings showed that increased CSF amyloid-beta 42 (Aβ42) levels correlated with decreased temporal rsEEG theta activity in ADD and increased global rsEEG alpha and beta power in both ADMCI and ADD patients. Furthermore, elevated CSF tau levels and higher p-tau/Aβ42 ratios were linked to increased global rsEEG theta and decreased alpha power. Finally, higher CSF Aβ42 levels were associated with increased global rsEEG delta activity in ADMCI patients, and negative associations were observed between the CSF p-tau/Aβ42 ratio and global rsEEG alpha activity. The variability in these findings may be attributable to the utilization of fixed frequency bands and methodological limitations, such as not accounting for head volume conduction effects [13].

In this retrospective study, we aimed to explore the relationship between the CSF p-tau/Aβ42 biomarkers typically used for the in vivo diagnosis of AD according to the NIA-AA Framework for AD diagnosis through biomarkers [1,2] and the topography and individual frequency bands of rsEEG rhythms in ADMCI patients compared to those with MCI not attributed to AD (noADMCI). Our Consortium’s experimental procedures enhanced the spatial resolution of rsEEG activity by estimating regional cortical source activity. Additionally, we utilized individually tailored frequency bands to account for the varying degrees of rsEEG slowing in ADMCI and noADMCI patients [23,24]. We hypothesized that rsEEG abnormalities would be more pronounced in ADMCI patients compared to noADMCI patients and that these abnormalities would correlate with CSF p-tau and Aβ42 biomarkers in the ADMCI group.

## 2. Results

### 2.1. Clinical, Genetic, and CSF Amyloid–Tau in noADMCI and ADMCI Groups

Table 1 summarizes the clinical (i.e., GDS, CDR, and HIS), genetic (i.e., APOE genotyping), and CSF amyloid–tau (i.e., Aβ42, t-tau, p-tau, and Aβ42/p-tau) markers in the noADMCI and ADMCI groups, together with the results of the statistical analyses computed to evaluate the presence or absence of statistically significant differences between these two groups regarding each clinical (T-test), genetic (Fisher test), and CSF amyloid–tau (T-test) variable. As expected, significant differences were found for the APOE genotyping (*p* < 0.00001) and the three CSF amyloid–tau markers (*p* < 0.00001). On the contrary, no statistically significant differences in the clinical markers were found (*p* > 0.05).

Table 2 reports (1) the mean values (±standard error of the mean, SE) of the following neuropsychological tests in the noADMCI and ADMCI groups: Logical Memory Test (immediate and delayed recall), Rey Auditory Verbal Learning Test (immediate and delayed recall), Trail Making Test part B-A, Verbal Fluency for letters, Verbal Fluency for category, clock drawing, and clock copy; (2) the cut-off scores of the above-mentioned neuropsychological tests; and (3) the results of the presence or absence of statistically significant differences (T-test; log−10 transformed data) between the noADMCI and ADMCI groups for the neuropsychological tests. The statistical threshold was set at *p* < 0.005 (i.e., 9 neuropsychological tests, *p* < 0.05/9 = 0.005) to obtain the Bonferroni correction at *p* < 0.05 and consider the inflating effects of repetitive univariate tests. No statistically significant differences were found (*p* > 0.005). Furthermore, a worsening in the Logical Memory Test (immediate recall, *p* = 0.03; delayed recall, *p* = 0.04) and Rey Auditory Verbal Learning Test (immediate recall, *p* = 0.02; delayed recall, *p* = 0.05) was found in the ADMCI group as compared to the noADMCI group using an explorative statistical threshold of *p*  <  0.05 uncorrected.

### 2.2. The rsEEG Source Activities in Healthy, noADMCI, and ADMCI Groups

The mean TF was 5.5 Hz (±0.2 SE) in the Healthy group, 5.5 Hz (±0.2 SE) in the noADMCI group, and 5.3 Hz (±0.1 SE) in the ADMCI group. Furthermore, the mean IAF was 9.2 Hz (±0.1 SE) in the Healthy group, 9.1 Hz (±0.2 SE) in the noADMCI group, and 8.8 Hz (±0.1 SE) in the ADMCI group. Two ANOVAs (*p* < 0.05) were performed to evaluate the presence or absence of statistically significant differences between the three groups regarding TF and IAF. No statistically significant differences were found (*p* > 0.05).

Figure 1 shows the mean values (± standard error of the mean, SE; log−10 transformed values) of rsEEG source activities (i.e., regional normalized eLORETA current densities) relative to a statistically significant ANOVA interaction effect (F = 6.0; *p* < 0.0001) among the factors Group (Healthy, noADMCI, and ADMCI), Band (delta, theta, alpha 1, alpha 2, alpha 3, beta 1, beta 2, and gamma), and ROI (frontal, central, parietal, occipital, and temporal).

The Duncan planned post hoc (*p* < 0.05, Bonferroni correction for 8 frequency bands X 5 ROIs, *p* < 0.05/40 = 0.00125) testing showed that the discriminant pattern Healthy > noADMCI > ADMCI was fitted by the parietal, occipital, and temporal (*p* < 0.00001) rsEEG alpha 2 source activities (*p* < 0.00001), as well as the parietal and temporal rsEEG alpha 3 source activities (*p* < 0.00001).

Furthermore, the Duncan planned post hoc (*p* < 0.00125) also showed the following effects: (1) the discriminant pattern Healthy > noADMCI and ADMCI was fitted by the central rsEEG alpha 2 and alpha 3 source activities (*p* < 0.00001); (2) the discriminant pattern Healthy, noADMCI > ADMCI was fitted by the occipital rsEEG alpha 3 source activities (*p* < 0.000005); (3) the discriminant pattern Healthy, noADMCI < ADMCI was fitted by the fontal rsEEG theta source activities (*p* < 0.0001); (4) the discriminant pattern Healthy < noADMCI, ADMCI was fitted by the parietal and occipital rsEEG delta source activities (*p* < 0.000001), as well as the occipital rsEEG theta source activities (*p* < 0.0001); and (5) the discriminant pattern Healthy < ADMCI was fitted by the central and temporal rsEEG delta source activities (*p* < 0.00001), as well as the central and parietal rsEEG theta source activities (*p* < 0.0001).

The findings mentioned above were not due to outliers from those individual regional normalized eLORETA current densities modeling rsEEG source activities (log−10 transformed), as shown by the Grubbs’ test with an arbitrary threshold of *p* > 0.001.

### 2.3. Associations Between CSF Amyloid–Tau Markers and rsEEG Source Activities in noADMCI and ADMCI Participants

General linear models assessed the association between three CSF amyloid–tau markers as predictors (i.e., CSF Aβ42, t-tau, and p-tau—log−10 transformed values) and five rsEEG source activities as dependent variables in the noADMCI, ADMCI, and extended MCI (i.e., ADMCI+noADMCI) groups. The rsEEG source activities were those showing statistically significant differences between the Healthy, noADMCI, and ADMCI groups (i.e., parietal alpha 2, occipital alpha 2, temporal alpha 2, parietal alpha 3, and temporal alpha 3 source activities; log−10 transformed values; see Table 3). A conservative statistical threshold at *p* < 0.0033 (i.e., Bonferroni correction at *p* < 0.05 for 2 CSF amyloid–tau markers X 5 rsEEG source activities, *p* < 0.05/15 = 0.0033) was set to consider the inflating effects of repetitive univariate tests.

In the ADMCI group, there were statistically significant negative associations between the CSF p-tau levels and the following rsEEG variables: the parietal rsEEG alpha 2 (β = −0.367, t = −3.25, *p* = 0.002), occipital rsEEG alpha 2 (β = −0.399, t = −3.58, *p* = 0.001), and parietal rsEEG alpha 3 (β = −0.345, t = −3.03, *p* = 0.003) source activities using the conservative statistical threshold of *p*  <  0.05 corrected (*p* < 0.0033). Using an explorative statistical threshold of *p*  <  0.05 uncorrected, the following associations were found: (1) a negative association between the CSF p-tau levels and the temporal rsEEG alpha 2 source activities (β = −0.284, t = −2.45, *p* = 0.01); and (2) negative associations between the CSF t-tau levels and the rsEEG parietal alpha 2 (β = −0.329, t = −2.87, *p* = 0.005), occipital alpha 2 (β = −0.298, t = −2.58, *p* = 0.01), temporal alpha 2 (β = −0.246, t = −2.09, *p* = 0.04), and parietal alpha 3 (β = −0.332, t = −2.90, *p* = 0.005) source activities. The greater the *p*-tau or CSF t-tau levels (increasing the probability of being positive to the diagnostic CSF biomarkers for AD), the lower the rsEEG alpha source activities. Figure 2 and Figure 3 show the scatterplots of those negative associations, showing the strict relationships between brain tauopathy and posterior rsEEG alpha rhythms in ADMCI patients.

In the noADMCI group, there was a statistically significant positive association between the CSF Aβ42 levels and the parietal rsEEG alpha 3 source activities (β = 0.436, t = 3.03, *p* = 0.003) using the conservative statistical threshold of *p*  <  0.05 corrected (*p* < 0.0033). Using an explorative statistical threshold of *p*  <  0.05 uncorrected, a positive association between the CSF Aβ42 levels and the parietal rsEEG alpha 2 source activities was also found (β = 0.362, t = 2.55, *p* = 0.01). The lower the CSF Aβ42 levels (increasing the probability of being positive to the diagnostic CSF biomarkers for AD), the lower the rsEEG alpha source activities. Figure 4 shows the scatterplots of those positive associations, showing a marginal relationship between brain amyloidosis and posterior rsEEG alpha rhythms in noADMCI patients.

In the extended MCI (i.e., ADMCI+noADMCI) group, there were statistically significant negative associations between the CSF p-tau levels and the parietal alpha 2 (β = −0.351, t = −3.98, *p* = 0.001), occipital alpha 2 (β = −0.387, t = −4.46, *p* = 0.001), temporal alpha 2 (β = −0.286, t = −3.17, *p* = 0.002), and parietal alpha 3 (β = −0.351, t = −3.99, *p* = 0.001) source activities estimated from the rsEEG activity using the conservative statistical threshold of *p*  <  0.05 corrected (*p* < 0.0033). Along the same line, statistically significant negative associations between the CSF t-tau levels and the parietal rsEEG alpha 2 (β = −0.323, t = −3.62, *p* = 0.001), occipital alpha 2 (β = −0.327, t = −3.67, *p* = 0.001), and parietal alpha 3 (β = −0.333, t = −3.75, *p* = 0.001) source activities were found. The higher the CSF p-tau or t-tau levels (increasing the probability of being positive to the diagnostic CSF biomarkers for AD), the lower the rsEEG alpha source activities.

Using an explorative statistical threshold of *p*  <  0.05 uncorrected, the following negative associations were also found in the extended MCI (i.e., ADMCI+noADMCI) group. There was a negative association between the CSF p-tau levels and the temporal rsEEG alpha 3 source activities (β = −0.267, t = −2.94, *p* = 0.004). Furthermore, there were negative associations between the CSF t-tau levels and the temporal rsEEG alpha 2 (β = −0.265, t = −2.92, *p* = 0.004) and alpha 3 (β = −0.266, t = −2.94, *p* = 0.004) source activities. Again, the greater the CSF p-tau or t-tau levels (increasing the probability of being positive to the diagnostic CSF biomarkers for AD), the lower the rsEEG alpha source activities.

Figure 5 and Figure 6 show the scatterplots of those negative associations, showing the strict relationships between the brain tauopathy and posterior rsEEG alpha rhythms in the extended MCI group.

Using an explorative statistical threshold of *p*  <  0.05 uncorrected, there were positive associations between the CSF Aβ42 levels and the parietal rsEEG alpha 3 (β = 0.224, t = 2.45, *p* = 0.01) and the temporal rsEEG alpha 3 (β = 0.183, t = 1.98, *p* = 0.05) source activities in the extended MCI (i.e., ADMCI+noADMCI) group. The lower the CSF Aβ42 levels (increasing the probability of being positive to the diagnostic CSF biomarkers for AD), the lower the rsEEG alpha source activities. Figure 7 shows the scatterplots of those positive associations, showing a marginal relationship between brain amyloidosis and posterior rsEEG alpha rhythms in the extended MCI group.

The findings mentioned above were not due to cerebrovascular lesions as shown by general linear models that used MRI-measured WM hypo-intensity as a covariate to assess the association between the three CSF amyloid–tau markers as predictors (i.e., CSF Aβ42, t-tau, and p-tau-log−10 transformed values) and the five relevant rsEEG alpha source activities as dependent variables (i.e., parietal alpha 2, occipital alpha 2, temporal alpha 2, parietal alpha 3, and temporal alpha 3 source activities; log−10 transformed values) in the ADMCI group. The statistical details of these results are reported in Table 4. Of note, no statistically significant difference (Mann–Whitney test, *p* > 0.05) was found in WM hypo-intensity between the noADMCI and ADMCI groups.

The findings mentioned above were also not due to outliers from those individual CSF amyloid–tau levels (log−10 transformed) and regional normalized eLORETA current densities (log−10 transformed) modeling rsEEG source activities, as shown by the Grubbs’ test with an arbitrary threshold of *p* > 0.001.

### 2.4. Control Analysis of the Structural MRI Markers in the noADMCI and ADMCI Groups

Table 5 reports the mean values (± SE) of the following structural MRI markers in the noADMCI group and in the ADMCI group: normalized total GM volume, normalized total WM volume, normalized hippocampus volume, normalized amygdala volume, mean cortical thickness, parietal cortical thickness, temporal cortical thickness, precuneus cortical thickness, and cuneus cortical thickness. Furthermore, Table 5 reports the results of the presence or absence of statistically significant differences (T-test; log−10 transformed data) between the noADMCI and ADMCI groups for the structural MRI markers mentioned above. The conservative statistical threshold was set at *p* < 0.0055 (i.e., Bonferroni correction at *p* < 0.05 for 9 statistical comparisons, *p* < 0.05/9 = 0.0055) to consider the inflating effects of repetitive univariate tests. The results are reported in the following. Parietal (*p* = 0.0002) and precuneus (*p* = 0.0008) cortical thicknesses were significantly lower in the ADMCI group than in the noADMCI group. Furthermore, using an explorative statistical threshold of *p*  <  0.05 uncorrected, the mean cortical thickness (*p* = 0.02), temporal cortical thickness (*p* = 0.01), and normalized hippocampus volume (*p* = 0.02) were lower in the ADMCI group as compared to the noADMCI group.

The findings mentioned above were not due to outliers from those individual MRI marker values (log−10 transformed), as shown by the Grubbs’ test with an arbitrary threshold of *p* > 0.001.

### 2.5. Control Analysis on Associations Between Structural MRI and CSF Aβ42, t-Tau, and p-Tau in noADMCI and ADMCI Participants

For control purposes, general linear models were used to assess the association between the three CSF amyloid–tau markers as predictors (i.e., CSF Aβ42, t-tau, and p-tau levels—log−10 transformed values) and the two MRI markers showing the statistically significant differences between the noADMCI and ADMCI groups. This analysis was performed in the ADMCI and noADMCI groups considered separately and in the extended MCI (i.e., ADMCI+noADMCI) group. A conservative statistical threshold was set at *p* < 0.00833 (i.e., Bonferroni correction at *p* < 0.05 for 3 CSF amyloid–tau markers X 2 MRI markers activities, *p* < 0.05/6 = 0.00833) to consider the inflating effects of repetitive univariate tests. The results are reported in Table 6.

In the noADMCI and ADMCI groups considered separately, no statistically significant association between the CSF amyloid–tau levels and the structural MRI markers was found using a statistical threshold of *p*  <  0.05 corrected (*p* < 0.00833). Using an explorative statistical threshold of *p*  <  0.05 uncorrected, a positive association between CSF Aβ42 levels and the parietal cortical thickness was found in the ADMCI group (β = 0.263, t = 2.17, *p* = 0.03). The lower the CSF Aβ42 levels (increasing the probability of positivity to AD-related diagnostic biomarkers for AD), the smaller the cortical thickness.

In the extended MCI (i.e., ADMCI+noADMCI) group, there were statistically significant negative associations between the CSF t-tau levels and the parietal (β = −0.293, t = −3.11, *p* = 0.002) and precuneus (β = −0.316, t = −3.38, *p* = 0.001) cortical thicknesses using a statistical threshold of *p*  <  0.05 corrected (*p* < 0.00833). The higher the CSF t-tau levels (increasing the probability of positivity to AD-related diagnostic biomarkers for AD), the lower the cortical thickness. Furthermore, there were statistically significant positive associations between CSF Aβ42 levels and the parietal (β = 0.428, t = 4.81, *p* = 0.001) and precuneus (β = 0.402, t = 4.45, *p* = 0.001) cortical thicknesses. The lower the CSF Aβ42 levels (increasing the probability of positivity to AD-related diagnostic biomarkers for AD), the smaller the cortical thickness.

Using an explorative statistical threshold of *p*  <  0.05 uncorrected, negative associations between the CSF p-tau levels and the parietal (β = −0.241, t = −2.52, *p* = 0.01) and precuneus (β = −0.233, t = −2.43, *p* = 0.01) cortical thicknesses were also found. The higher the CSF p-tau levels (increasing the probability of positivity to AD-related diagnostic biomarkers for AD), the smaller the cortical thickness.

The findings mentioned above were not due to outliers from those individual CSF amyloid–tau levels (log−10 transformed) and regional normalized eLORETA current densities (log−10 transformed), as shown by the Grubbs’ test with an arbitrary threshold of *p* > 0.001.

### 2.6. Control Analysis of Associations Between Structural MRI and rsEEG Markers in noADMCI and ADMCI Participants

For control purposes, general linear models assessed the association between relevant structural MRI and rsEEG markers in the noADMCI, ADMCI, and extended MCI (i.e., ADMCI+noADMCI) groups. These markers were those showing statistically significant differences between the Healthy, noADMCI, and ADMCI groups in the above statistical comparisons. Specifically, we used parietal and precuneus cortical thicknesses (log−10 transformed values) as structural MRI markers, and parietal alpha 2, occipital alpha 2, temporal alpha 2, parietal alpha 3, and temporal alpha 3 source activities derived the rsEEG rhythms (log−10 transformed values). A conservative statistical threshold was set at *p* < 0= 0.005 (i.e., Bonferroni correction at *p* < 0.05 for 2 MRI neurodegeneration markers X 5 rsEEG source activities, *p* < 0.05/10 = 0.005) to consider the inflating effects of repetitive univariate tests. The results are reported in Table 7.

In the noADMCI and ADMCI groups considered separately, no statistically significant association between the structural MRI and rsEEG markers was found using the conservative statistical threshold of *p*  <  0.05 corrected (*p* < 0.005).

Using an explorative statistical threshold of *p*  <  0.05 uncorrected, we found positive associations between the parietal cortical thicknesses and the following rsEEG markers in the noADMCI group: the temporal alpha 2 (β = 0.317, t = 2.10, *p* = 0.04), parietal alpha 3 (β = 0.335, t = 2.19, *p* = 0.03), and temporal alpha 3 (β = 0.326, t = 2.13, *p* = 0.04) source activities. There were also positive associations between the precuneus cortical thicknesses and the following rsEEG markers: the temporal alpha 2 (β = 0.345, t = 2.27, *p* = 0.03), parietal alpha 3 (β = 0.326, t = 2.13, *p* = 0.04), and temporal alpha 3 (β = 0.333, t = 2.18, *p* = 0.03) source activities. The smaller the cortical thickness, the lower the rsEEG alpha source activities.

In the extended MCI (i.e., ADMCI+noADMCI) group, a statistically significant positive association (β = 0.275, t = 2.9, *p* = 0.005) between the parietal cortical thicknesses and the parietal rsEEG alpha 3 source activities was found using a statistical threshold of *p*  <  0.05 corrected (*p* < 0.005). Using an explorative statistical threshold of *p*  <  0.05 uncorrected, there were positive associations between the parietal cortical thicknesses and the parietal rsEEG alpha 2 (β = 0.213, t = 2.21, *p* = 0.02) and the temporal rsEEG alpha 3 (β = 0.197, t = 2.1, *p* = 0.04) source activities. There were also positive associations between the precuneus cortical thicknesses and the following rsEEG markers: parietal alpha 2 (β = 0.223, t = 2.32, *p* = 0.02), parietal alpha 3 (β = 0.265, t = 2.7, *p* = 0.006), and temporal alpha 3 (β = 0.201, t = 2.1, *p* = 0.04) source activities. The smaller the cortical thickness, the lower the rsEEG alpha source activities.

The findings mentioned above were not due to outliers from those individual MRI markers (log−10 transformed) and rsEEG source activities estimated by regional normalized eLORETA current densities (log−10 transformed), as shown by the Grubbs’ test with an arbitrary threshold of *p* > 0.001.

## 3. Discussion

In this exploratory study, we tested whether the abnormalities of cortical source activities of rsEEG delta, theta, and alpha rhythms, possibly underpinning the quiet vigilance dysregulation, may be greater in ADMCI patients than in noADMCI patients and may be associated with diagnostic CSF Aβ42 and p-tau biomarkers in the ADMCI patients at the individual level. Those biofluid biomarkers are typically used for the in vivo diagnosis of AD according to the NIA-AA Framework for AD diagnosis through biomarkers [1,2].

The results showed that rsEEG alpha source activities in the parietal, temporal, and occipital regions were significantly reduced in the ADMCI group compared to both the Healthy control group and the matched MCI group not due to AD (noADMCI), the neurobiological AD diagnosis being determined by the standard diagnostic CSF Aβ42 and p-tau biomarkers of the NIA-AA Framework in the use of biomarkers in AD [1,2]. Additionally, cortical atrophy in the parietal, temporal, and precuneus regions was more pronounced in the ADMCI group than in the noADMCI group. These findings corroborate previous evidence from our international Consortium, which employed the same methodology. Specifically, rsEEG alpha source activities in posterior cortical regions (parietal, temporal, and occipital) using individualized alpha frequency banding were particularly associated with AD, even in its prodromal stage of amnestic MCI, compared to other neurodegenerative diseases [23,24].

As novel results of the present study, there was a negative association between the CSF p-tau and t-tau levels and the posterior rsEEG alpha (but not delta–theta) source activities across the ADMCI individuals and all MCI (i.e., ADMCI+noADMCI) individuals enrolled in the present study. In contrast, only a marginal positive association between the CSF Aβ42 levels and the posterior rsEEG alpha source activities was observed across the noADMCI individuals and all MCI (i.e., ADMCI+noADMCI) individuals. These results indicate that in the prodromal stages of Alzheimer’s disease (ADMCI), abnormalities in posterior rsEEG alpha rhythms are more closely associated with brain tauopathy and neurodegeneration than with amyloidosis, based on current CSF diagnostic biomarkers and a relatively small sample size. This finding highlights the stronger relationship between brain tauopathy and rsEEG alpha rhythms compared to previous studies that linked both amyloid and tau biomarkers in the brain to abnormalities across various rsEEG rhythms, from delta to beta bands, in ADD and ADMCI patients [15,16,17,18,19,20,21,22]. Specifically, earlier studies showed a significant association between CSF Aβ42 levels and rsEEG markers, such as temporal theta rhythms in ADD patients [18] and global delta, alpha, and beta rhythms in ADMCI and ADD patients [17,20]. They also demonstrated a strong link between CSF t-tau, p-tau, and the p-tau/Aβ42 ratio with global rsEEG theta and alpha rhythms [15,19,20,22]. Considering these findings alongside the current results, we propose that the use of our rsEEG methodology (involving source topography and individual alpha frequencies) may help to reduce the variability in the observed effects of AD neuropathology on the topography of rsEEG delta, theta, and alpha rhythms, particularly during the prodromal stages of AD [13].

### 3.1. A Neurophysiological Model of Posterior rsEEG Alpha Rhythms in Humans a Century After Hans Berger Discovered Human EEG

Hans Berger discovered human rsEEG activity in a 17-year-old boy at his Psychiatric Clinic in Jena, Germany, on 6 July 1924. He used a bipolar montage of two electrodes into a skull breach. In 1929, Berger published his first article showing rsEEG traces and emphasized the presence of rsEEG alpha and beta rhythms [25].

A century later, posterior rsEEG alpha rhythms in healthy adults are widely recognized as reflecting the regulation of the neuromodulatory subcortical ascending systems involved in cortical arousal and vigilance during quiet wakefulness [11,26]. In support of this, previous studies have demonstrated a reduction in the amplitude of posterior rsEEG alpha rhythms following sleep deprivation [27], and a negative correlation between alpha rhythm amplitude and skin conductance levels reflecting autonomic arousal [28]. In another study, a brief transcranial vagal nerve stimulation in healthy adults (relative to sham stimulations) caused transient pupil dilation and attenuation of occipital rsEEG alpha rhythms, consistent with the known effects of such a stimulation on the nucleus tractus solitaris in the brainstem and, consequently, the locus coeruleus nucleus, a part of the subcortical arousal system [29,30]. Moreover, inhibiting the occipital cortex through transcranial static magnetic field stimulations led to a localized increase in the rsEEG alpha rhythms that signify cortical inhibition and a reduction in visual search performance during a separate session [31]. Furthermore, short transcranial magnetic stimulations on the dorsal premotor cortex produced less responses, evidenced by a blood oxygen level-dependent (BOLD) signal, observed in a resting-state functional MRI (rsfMRI) across the bilateral cortico-subcortical (striatum–thalamus) motor systems, when released during periods of ample rsEEG alpha rhythms [32]. Finally, a positive association between rsEEG alpha rhythms and the BOLD activity in the thalamus, along with a primarily negative association with BOLD activity in the visual and attentional posterior cerebral area, has been observed during quiet wakefulness [33,34,35,36]. These findings collectively support a neurophysiological model in which rsEEG alpha rhythms reflect cortical inhibition predominantly driven by thalamocortical signaling, thereby modulating cortical arousal and vigilance/consciousness level in humans.

In this context, the present results in prodromal AD patients unveil an association between AD-related tau neuropathology, parietal cortical neurodegeneration, and the neurophysiological oscillatory mechanisms generating parietal and occipital rsEEG alpha rhythms. Considering the low spatial resolution of the present EEG approach, we can only speculate about the localization of the EEG effects and the clinical implications related to such an association.

The brain is affected by various slow, gradual, morphological changes during physiological aging, including cerebral atrophy, alterations in gray and white matter, volume loss, ventricular enlargement, and widening of the sulci [37]. While the volume of the frontal and temporal lobes decreases, the occipital and parietal lobes show negligible age-related changes in volume [38]. In contrast, AD and related neurodegenerative diseases induce relatively fast and progressive changes in encephalic structures and the related neural networks, even before the disease’s clinical manifestations. They include early atrophy and gray matter (GM) loss in the subcortical neuromodulatory ascending systems regulating cerebral arousal and vigilance [39] and changes in the hippocampus, medial temporal lobe, and parietal lobe [40,41,42].

In AD-related posterior cortical atrophy, patients experience difficulties in driving, recognizing objects in pictures, and other visual deficits in relation to dysfunctions in the parietal and occipital areas and ascending cholinergic systems, which regulate visual and visuospatial–somatomotor information processing [40,41,42]. In the parietal cortex, the main targets of AD-related neuropathology and neurodegeneration are the precuneus and regions around the lateral temporoparietal junctions [43,44,45]. Among them, the precuneus is considered an important posterior node of the default mode network that processes somatic milieu, self-reflection conscious processes (comprising mind wandering), episodic/autobiographical memory retrieval, and switches to bottom-up sensory information processing [46,47,48].

In the above framework, the present evidence of abnormalities in CSF tauopathy, parietal cortical volume, and parietal rsEEG alpha source activity in ADMCI patients lends support to the idea of a substantial inhibitory/excitatory imbalance in the parietal and occipital cortical areas and vigilance regulation in prodromal AD. Along this speculative line, a transcranial magnetic stimulation over the precuneus in ADD patients induced locally enhanced EEG-evoked responses, suggesting local cortical hyperexcitability [49]. There was also a lack of the propagation of those responses in relation to abnormalities in the functional connectivity within the default mode network, as revealed by functional MRI measurements [49]. Furthermore, an effective transcranial magnetic stimulation at 20 Hz over the precuneus for 24 weeks was safe and slowed down cognitive and functional decline in ADD patients [50]. Another study is in progress to evaluate the safety, feasibility, and clinical effects of transcranial electric stimulation at 40 Hz over the precuneus for 8 weeks at ADD patients’ homes [51].

### 3.2. A Tentative Model Linking Neurobiology and Clinical Neurophysiology in Prodromal AD Stages

At the present stage, we can only speculate on the clinical neurophysiological model explaining the present finding of a remarkable relationship between brain tauopathy as revealed by CSF diagnostic biomarkers and posterior rsEEG alpha rhythms in the prodromal AD stage of amnesic MCI.

Tau protein plays several roles at the level of axon microtubules and synapsis. It can intracellularly accumulate in the brain neurons of AD patients as neurofibrillary tangles and may propagate through the brain in a prion-like manner in relation to neuronal activity [52]. Therefore, it may pathophysiologically interfere with neuromodulatory subcortical systems in AD patients and induce abnormalities in cortical neural arousal, as revealed by posterior rsEEG alpha rhythms [39].

Along this speculative line, previous studies showed that tau-related degeneration within neuromodulatory subcortical systems can be found in AD patients at earlier disease stages, even before observable memory deficits and volumetric cortical atrophy (for a recent review, see [39]). In the disease course, such a degeneration may also have a major role in the derangement of those systems and the induction of significant behavioral and cognitive deficits [39]. Among its pathophysiological actions, previous studies showed neuroinflammation and interferences with synaptic neurotransmission, the temporal synchronization of cortical neural activity and connectivity, and the cortical excitatory/inhibitory balance as reflected in the generation of brain rhythms [39,53,54]. Other studies reported that tau protein spreads and the related hypometabolism in the brain from PET data were associated with abnormal resting-state magnetoencephalographic (rsMEG) activity, including a reduced power in posterior alpha rhythms measured in ADD patients [55,56]. Furthermore, neurofibrillary tangle density was evaluated in an autopsy examination of ADD patients at a post-mortem neuropathological examination, and its measurement was predicted by abnormal rsMEG alpha (but not delta–theta) rhythms [57]. Finally, brain tau deposition from PET data was associated with a computational measurement reflecting abnormally increased cortical neural excitation in ADD patients, which conceptually fit with a reduction in posterior rsEEG alpha rhythms [11,58].

Notably, the present findings in the ADMCI and the extended group of MCI patients with and without AD neuropathology patients showed a more significant association between posterior rsEEG alpha rhythms and CSF tau levels when compared to CSF aβ42/40 levels. The marginal association between the posterior rsEEG alpha rhythms and CSF aβ42/40 levels may be due to the relatively low number of ADMCI patients and the well-known more frontal distribution of cortical amyloidosis at the earlier stages of AD. Future research with combined amyPET and rsEEG recordings should further investigate this speculative explanation.

### 3.3. Methodological Remarks

In this exploratory study, clinical and rsEEG datasets were obtained from several clinical units, not all of which underwent a preliminary rigorous phase of standardizing operating procedures for data collection in a prospective clinical trial.

The current study employed the 10–20 montage system, utilizing 19 scalp electrodes for rsEEG recordings. This electrode configuration is considered appropriate for exploratory retrospective rsEEG studies involving ADMCI/noADMCI patients, especially when using source estimation techniques at a low spatial resolution [13]. Due to this limitation, the cortical sources of rsEEG rhythms were estimated within broad cortical regions of interest (i.e., cortical lobes) instead of a fine source localization. Additionally, we applied the eLORETA source estimation method, which is particularly effective for modeling spatially widespread cortical source activations, thanks to its smoothing regulation procedures [59,60,61]. While an eLORETA source estimation using rsEEG recordings from 30 scalp electrodes can yield informative preliminary results, future studies should use high-resolution EEG techniques with 64–256 scalp electrodes to reach a high spatial resolution in the rsEEG source estimation [62,63].

The present methodological choice also allowed us to compare the current results with those of our previous studies following the same methodology in patients with MCI due to not only AD but also Parkinson’s and Lewy body diseases [23,24]. It essentially probes the local cortical synchronization/desynchronization of the activity of large cortical neural populations at delta, theta, and alpha frequencies. The study results showed that AD-related tau neuropathology significantly affects such activity in ADMCI patients at the low spatial resolution allowed by the general methodology and encourages the modeling of the underlying functional cortical connectivity in future studies using EEG recordings with a higher number of scalp electrodes (>30 electrodes).

Finally, the experimental design did not include external stimuli and cognitive–motor demands to provide behavioral measures of vigilance to be correlated with the EEG data. At this early research stage, these demands were not included to avoid interferences with the scope of the resting-state condition (e.g., induce spontaneous EEG activity).

## 4. Materials and Methods

### 4.1. Participants

The datasets for the present study were obtained from the international PharmaCog and PDWAVES Consortium (www.pdwaves.eu) archives. They included records from demographic-matched groups (i.e., the groups had the same mean values of age, gender, and sex ratio) consisting of 70 ADMCI, 45 noADMCI, and 45 Healthy participants, all of whom underwent rsEEG recordings under the eyes-closed condition. Participants were recruited from various clinical centers, including the Sapienza University of Rome (Italy), Institute for Research and Evidence-based Care (IRCCS) “Fatebenefratelli” of Brescia (Italy), IRCCS SDN of Naples (Italy), IRCCS Oasi Maria SS of Troina (Italy), IRCCS Ospedale Policlinico San Martino and DINOGMI (University of Genova, Italy), Hospital San Raffaele of Cassino (Italy), Hospital of Perugia (Italy), Hospital of Chieti (Italy), IRCCS San Raffaele Pisana of Rome (Italy), Medipol University of Istanbul (Turkey), and Dokuz Eylül University of Izmir (Turkey).

Table 8 provides a summary of the relevant demographic (i.e., age, sex, and education) and clinical (i.e., MMSE score) characteristics of the Healthy, noADMCI, and ADMCI groups, along with the results of the statistical analyses computed to determinate the presence or absence of statistically significant differences between these groups in terms of age (ANOVA), sex (Freeman–Halton test), education (ANOVA), and MMSE score (Kruskal–Wallis test). As anticipated, significant differences in MMSE scores were observed between the Healthy group and both the noADMCI and ADMCI groups (H = 39.1, *p* < 0.0001), with the Healthy group showing higher scores (*p* < 0.00001). In contrast, there were no statistically significant differences in age, sex, and education among the three groups (*p* > 0.05).

The study received approval from a local institutional ethics committee (see Appendix A, Ethical approval statement for details of the name of the local institutional ethics committee and the approval code of each unit). All procedures were carried out with informed consent obtained openly from each participant or their caregiver, adhering to the ethical guidelines of the World Medical Association (Declaration of Helsinki) and the standards set by the local institutional review boards.

### 4.2. Diagnostic Criteria

The clinical inclusion criteria for both ADMCI and noADMCI participants were as follows: (1) age range of 55 to 90 years; (2) self-reported memory concerns by the participant; (3) Mini-Mental State Examination (MMSE) score of 24 or higher; (4) Clinical Dementia Rating (CDR) score of 0.5 [64]; (5) Logical Memory Test [65] performance of 1.5 standard deviations (SDs) below the age-adjusted mean, indicating cognitive impairment that does not significantly affect functional independence in daily activities; (6) Geriatric Depression Scale (15-item GDS) score of 5 or lower [66]; (7) Modified Hachinski Ischemia score of 4 or lower [67]; (8) at least 5 years of education; and (9) diagnosis of single- or multi-domain MCI. Notably, individuals with a history of major depressive episodes or a prior diagnosis of major depressive disorder or other significant psychopathologies were excluded based on their self-reports during an initial clinical interview conducted by a licensed clinical psychologist to evaluate current and past mental health issues.

The clinical exclusion criteria for both ADMCI and noADMCI groups included the following: (1) mixed dementia; (2) chronic use of neuroleptics, narcotics, analgesics, sedatives, or hypnotics (e.g., benzodiazepines); (3) ongoing participation in a clinical trial involving disease-modifying drugs; (4) diagnosis of major psychiatric disorders (i.e., depression, etc.) or neurological illness not related to cognitive deficits; (5) diagnosis of epilepsy or report of seizures or epileptiform EEG activity in the past; and (6) use of antiepileptics.

The clinical status of ADMCI was determined based on positivity to CSF core AD markers, as reported below, associated with a compatible neurodegeneration pattern at structural MRI or FDG-PET [7]. All noADMCI subjects, conversely, were CSF-negative.

Both ADMCI and noADMCI participants underwent a comprehensive cognitive assessment, which included the following: (1) global cognitive function was assessed using the Mini-Mental State Examination (MMSE) score [68]; (2) episodic memory was evaluated with immediate and delayed recall tasks of the Logical Memory and the Rey Auditory Verbal Learning Test [65,69]; (3) executive functions and attention were assessed using the Trail Making Test (TMT) parts A and B [70]; (4) language abilities were measured using the 1 min Verbal Fluency Test for letters and categories [71]; and (4) planning abilities and visuospatial skills were evaluated with the Clock Drawing and Copy Test [72].

Healthy participants underwent cognitive screening, including MMSE and Geriatric Depression Scale, as well as physical and neurological examinations to rule out subjective memory complaints, cognitive deficits, and mood disorders. Exclusion criteria for healthy controls were (1) past or present neurological or psychiatric disorders, (2) depressive symptoms identified by a Geriatric Depression Scale (15-item version) score higher than 5, (3) chronic use of psychoactive drugs, and (4) presence of significant chronic systemic illnesses such as diabetes mellitus.

### 4.3. Cerebrospinal Fluid (CSF) Diagnostic Biomarkers

CSF diagnostic biomarkers for AD were assessed in all ADMCI and noADMCI participants. The CSF samples were preprocessed, frozen, and stored following the Alzheimer’s Association Quality Control Program for CSF biomarkers [73]. Levels of amyloid beta 1–42 (i.e., Aβ42), total tau (i.e., t-tau), and phosphorylated tau at residue 181 (i.e., p-tau) were measured using dedicated single-parameter colorimetric enzyme-linked immunosorbent assay (ELISA) kits (Innogenetics, Ghent, Belgium). Assays were performed in parallel from a single frozen aliquot of CSF, according to the manufacturer’s instructions, with each sample being tested in duplicate. A sigmoidal standard curve was created to quantify the light absorbance (pg mL^−1^). All ADMCI participants in this study were “positive” to the CSF Aβ42/p-tau biomarker, with a threshold defined in a previous investigation of our Workgroup [74]. In that investigation, the positivity cut-off for the CSF Aβ42/p-tau ratio was 15.2 for APOE4 carriers and 8.9 for non-carriers. Accordingly, in this study, all ADMCI participants with APOE4 status had a CSF Aβ42/p-tau lower than 15.2, whereas those without APOE4 had a ratio below 8.9.

### 4.4. Magnetic Resonance Imaging (MRI)

MRI scans were conducted for both ADMCI and noADMCI participants with different MRI systems (Siemens, GE, and Philips) using different fields (1.5 and 3T). The MRI protocol included anatomical scans using T1-weighted imaging. The MRI protocols were standardized, and optimization of the pipelines was carried out. Each session included multiple acquisitions using only vendor-provided sequences, such as anatomical T2*, anatomical FLAIR, resting-state fMRI, B0 map, DTI, and two anatomical T1 scans, all performed without repositioning the subject. The total mean acquisition time was approximately 40 min. The DTI scan was always the final acquisition, regardless of the MRI site. MRI data were formatted according to Brain Imaging Data Structure (BIDS) standards to ensure consistency and compatibility. Default geometric distortion correction settings were used for each scanner. Multi-channel coil images were reconstructed by summing squares across channels. When supported by the MRI system, images were saved without additional filtering to avoid introducing variability in smoothing across scanners. The 3T data included 81% of ADMCI and 87% of noADMCI patients, with a statistical analysis showing no significant differences. The acquired MRI data were anonymized according to international standards to ensure the protection of sensitive biomedical information. The analysis was centrally conducted by the research group at the Sapienza University of Rome, and each piece of data underwent visual inspection for quality assurance (i.e., visible artifacts including head motion, wrap-around, radio frequency interference, and signal intensity or contrast inhomogeneities) before further analyses.

Volumetric and cortical thickness MRI markers were obtained using the following approach. T1-weighted images were averaged within each session, and anatomical scans were subsequently processed using FreeSurfer (Dale, 1999) to automatically generate estimates of cortical thickness and subcortical volume for each participant [75,76,77] in predefined regions of interest (ROIs). Our study focused on a subset of MRI markers relevant to neurodegenerative diseases. These MRI markers included the following: (1) total gray matter (GM) and white matter (WM) volumes, normalized to total intracranial volume; (2) total cortical thickness; (3) hippocampus and amygdala volumes, normalized to the total intracranial volume; and (4) thickness measurements of specific cortical regions, including the parietal cortex, temporal cortex, precuneus, and cuneus. Segmentation results were visually inspected before proceeding with volume and thickness analyses, and no manual adjustment was made.

For subcortical white matter, FreeSurfer identified hypodense areas by detecting regions of lower intensity within the segmented white matter. These areas can be indicative of cerebrovascular lesions, such as small vessel disease, or chronic ischemia, like T2-weighted image white matter hyperintensities in clinical contexts [78]. The software v6.0 computes the volume or spatial extent of these hypodense regions and integrates these data into an index or measure of cerebrovascular damage. The subcortical white matter hypodensity measure was to evaluate the control hypothesis of a greater cerebrovascular burden in the noADMCI group than in the ADMCI group by the Mann–Whitney test (*p* < 0.05). Furthermore, it was used as a covariate for investigating the association between rsEEG source activities and CSF diagnostic biomarkers in the ADMCI group.

### 4.5. rsEEG Recordings

The rsEEG recordings were conducted using local routine professional digital EEG systems licensed for clinical applications. The following digital EEG systems were used: BrainAmp 32-Channel DC System (Brain Product GmbH, Gilching, Germany), Waveguard caps (ANT Neuro, Hengelo, The Netherlands), EB Neuro-BE LIGHT (EB Neuro, Florence, Italy), Galileo NT Line—EB Neuro (EB Neuro, Italy), and EB Neuro-Sirius BB (EB Neuro, Italy). rsEEG recordings were performed in all participants using at least 19 scalp exploring electrodes (i.e., 19, 32, or 64 channels), placed according to the 10–20 system. The selected electrode montage included the 19 scalp monopolar channels placed following the 10–20 System (i.e., Fp1, Fp2, F7, F3, Fz, F4, F8, T7, C3, Cz, C4, T8, P7, P3, Pz, P4, P8, O1, and O2). All rsEEG recordings were performed in the morning to minimize potential variations due to circadian rhythms. Standard instructions for the resting-state condition emphasized staying awake, psychophysically relaxed with mind wandering, and following the experimenter’s requests to keep the eyes closed and open during the rsEEG recording. The experiments checked the participant’s behavioral state during the EEG recordings and annotated eventual deviations and alarms.

A common electrode montage of 19 scalp exploring electrodes, placed according to the 10–20 system (i.e., O1, O2, P3, Pz, P4, T3, T5, T4, T6, C3, Cz, C4, F7, F3, Fz, F4, F8, Fp1, and Fp2), characterized the EEG data recordings in all clinical units and was used for the data analysis. The reference electrode was typically placed between Fz and Cz of the 10–20 system, and the ground electrode was in the posterior midline. To minimize the influence of the different placement of the reference electrodes, all EEG data were re-referenced to the common average for the data analysis.

Electrooculographic (EOG) activity with a standard bipolar montage was also recorded to monitor and control eye movements and blinking. As minimum standards in all clinical units, the electrophysiological data allowed a bandpass filtering of 0.3–70 Hz and a sampling rate of 256 Hz.

### 4.6. Preliminary rsEEG Data Analysis

The rsEEG data were centrally analyzed by experts at the Sapienza University of Rome, who were blinded to the participant’s diagnosis. The recorded rsEEG data were exported in either European data format (.edf) or EEGLAB set (.set) files and subsequently processed offline using the EEGLAB toolbox [79] (version eeglab14_1_2b) running in MATLAB software (Mathworks, Natick, MA, USA; version: R2014b).

For preprocessing, the rsEEG data were segmented into 2 s epochs (i.e., 5 min of data corresponded to 150 epochs of 2 s each) and subjected to offline analysis. A three-step procedure was implemented to detect and remove the following items: (1) recording channels (electrodes) showing prolonged artifactual rsEEG activity due to bad electric contacts or other reasons; (2) rsEEG epochs containing artifacts from channels that generally had good signals; and (3) intrinsic components of the rsEEG epochs affected by artifacts.

The initial step involved a visual examination of rsEEG activity by two independent experimenters from a panel of four experts (i.e., C.D.P, R.L., S.L., and G.N.) to identify electrodes with irreparable artifacts, typically removing no more than three per participant. If a clinical unit used a digital EEG system with more than 19 exploring electrodes, the removed electrodes were substituted with the nearest non-selected ones. These additional electrodes, along with artifact-free ones, were used to interpolate data at the locations of the removed electrodes, ensuring all participants had artifact-free EEG data.

In the second step, the same experimenters visually reviewed the rsEEG epochs to identify and eliminate those contaminated by muscular, ocular, or head movements or non-physiological artifacts. Muscle tension artifacts were detected by examining power density spectra, revealing unusually high values in the 30 to 70 Hz range, which deviated from the typical decline in power density.

The third step involved applying an independent component analysis (ICA) using the EEGLAB toolbox to remove components representing residual artifacts, including eye movements, involuntary head motions, neck and shoulder muscle tension, and electrocardiographic activity [80,81]. Fewer than 3 ICA components were removed from each dataset, which were reconstructed with the remaining artifact-free ICA components. To ensure data integrity, the presumed artifact-free rsEEG epochs underwent a visual double check by the independent experimenters, confirming their inclusion or exclusion.

To harmonize the data, the artifact-free EEG recordings for the common 19 electrodes underwent digital frequency-band passing at 0.1–45 Hz. When necessary, the data were down-sampled to ensure a uniform sampling rate of 256 Hz across all artifact-free rsEEG datasets. Additionally, the EEG data were re-referenced to the common average reference.

After these procedures, the artifact-free epochs maintained a similar proportion (over 75%) of the total rsEEG activity recorded across all participants (i.e., ADMCI, noADMCI, and Healthy).

### 4.7. Spectral Analysis of the rsEEG Epochs

The standard digital Fast Fourier Transform (FFT) analysis, using the Welch technique with a Hanning window and no phase shift, was applied to compute the power density of the artifact-free rsEEG epochs recorded at all 19 scalp electrodes, with a frequency resolution of 0.5 Hz.

The rsEEG frequency bands of interest were defined using specific frequency landmarks: the transition frequency (TF) and the individual alpha frequency (IAF). In the (eyes-closed) rsEEG power density spectrum, the TF was defined as the minimum rsEEG power density between 3 and 8 Hz, while the IAF peak was defined as the maximum power density peak between 6 and 14 Hz. Both TF and IAF were calculated for each participant.

Based on these landmarks, individual delta, theta, and alpha bands were estimated as follows: delta from TF −4 Hz to TF −2 Hz, theta from TF −2 Hz to TF, low alpha (alpha 1 and alpha 2) from TF to IAF, and high-frequency alpha (or alpha 3) from IAF to IAF + 2 Hz. Specifically, individual alpha 1s ranged from the TF to the midpoint of the TF-IAF range, while alpha 2 extended from that midpoint to the IAF peak. The remaining bands were defined based on standard fixed frequency ranges from reference rsEEG studies [74,82,83,84,85]: beta 1 from 14 to 20 Hz, beta 2 from 20 to 30 Hz, and gamma from 30 to 40 Hz.

### 4.8. Estimation of rsEEG Source Activation

We used the freeware tool exact LORETA (eLORETA) to linearly estimate the cortical source activity generating scalp-recorded rsEEG rhythms [59]. The present eLORETA implementation uses a head volume conductor model comprising the scalp, skull, and cerebral cortex. The electrodes can be virtually positioned in the scalp compartment to provide EEG data for source estimation [59]. The cortical model is based on the Montreal Neurological Institute (MNI152 template), allowing the solution of the EEG inverse problem and estimation of “neural” current density values at any cortical voxel within the model.

To estimate EEG cortical source activities (i.e., eLORETA solutions), we used spectral power density data from the 19 scalp electrodes. This process occurs within the electrical cortical source space, which includes 6239 voxels with a 5 mm resolution, limited to the cortical gray matter of the head volume conductor model. Each voxel contains an equivalent current dipole representing the mean ionic currents from local populations of cortical pyramidal neurons. The eLORETA package provides Talairach coordinates, lobe, and Brodmann area (BA) for each voxel.

Normalization of the eLORETA solutions was performed by averaging across all frequency bins (0.5 to 45 Hz) and the 6239 voxels to obtain the eLORETA “mean” solution. We then computed the ratio between each original eLORETA solution at a specific frequency bin/voxel to the eLORETA mean solution, resulting in a normalized eLORETA.

Given the low spatial resolution of our EEG methodology (i.e., 19 scalp electrodes), we conducted a regional analysis of the eLORETA solutions. This involved collapsing the eLORETA solutions within frontal (BAs 8, 9, 10, 11, 44, 45, 46, and 47), central (BAs 1, 2, 3, 4, and 6), parietal (BAs 5, 7, 30, 39, 40, and 43), occipital (BAs 17, 18, and 19), and temporal (BAs 20, 21, 22, 37, 38, 41, and 42) macro-regions (ROIs).

For the present eLORETA cortical source estimation, a frequency resolution of 0.5 Hz was used.

### 4.9. Main Statistical Analysis of rsEEG Source Activities

Two main statistical analyses were conducted to test the two working study hypotheses.

The first statistical analysis was conducted using the commercial software STATISTICA 10 (StatSoft Inc., 2300 E 14th St Tulsa OK Oklahoma, United States 74104, www.statsoft.com) to test the first working hypothesis that the rsEEG source activities may differ among the ADMCI, noADMCI, and Healthy groups. An ANOVA was performed, with rsEEG source activities (i.e., regional normalized eLORETA solutions) as a dependent variable (*p* < 0.05). The ANOVAs included the factors Group (Healthy, noADMCI, and ADMCI), Band (delta, theta, alpha 1, alpha 2, alpha 3, beta 1, beta 2, and gamma), and ROI (frontal, central, parietal, occipital, and temporal). Post hoc comparisons were conducted using the Duncan test (*p* < 0.05, Bonferroni corrected). Confirmation of the first working hypothesis required two criteria: (1) a statistically significant ANOVA interaction involving the Group factor (*p* < 0.05) and (2) a post hoc Duncan test revealing statistically significant (*p* < 0.05, Bonferroni corrected) differences in the rsEEG source activities between the three groups (Healthy ≠ noADMCI ≠ ADMCI).

The second statistical analysis was conducted using the freeware tool Jamovi Version 2.3 (www.jamovi.org) to test the second working hypothesis that the rsEEG source activities may be associated with the CSF amyloid–tau markers in the ADMCI and noADMCI patients. For the ADMCI, noADMCI, and extended MCI (i.e., ADMCI+noADMCI) groups, several general linear models (*p* < 0.05, Bonferroni corrected) were implemented, namely one model for each rsEEG showing significant differences between the ADMCI and noADMCI groups in the above statistical analysis and CSF markers as predictors (i.e., CSF Aβ42, t-tau, and p-tau).

The potential impact of rsEEG marker outliers on the statistical results was evaluated using the iterative (leave-one-out) Grubbs’ test to detect outliers. The null hypothesis of non-outlier status was tested at a threshold of *p* > 0.001 to remove individual values with a high probability of being outliers.

### 4.10. Control Statistical Analysis of MRI Markers

Three control statistical analyses were conducted to test the two control study hypotheses.

The first control statistical analysis was conducted using the commercial software STATISTICA 10 to test the first control hypothesis that the MRI neurodegeneration markers may differ between the ADMCI and noADMCI groups. Several T-tests were performed (*p* < 0.05, Bonferroni corrected) using various MRI markers as dependent variables: normalized total GM volume, normalized total WM volume, normalized hippocampus volume, normalized amygdala volume, mean cortical thickness, parietal cortical thickness, temporal cortical thickness, precuneus cortical thickness, and cuneus cortical thickness.

The second control statistical analysis was conducted using the freeware tool Jamovi Version 2.3 to test whether MRI neurodegeneration markers may be associated with CSF amyloid–tau markers in the ADMCI and noADMCI patients. For the ADMCI, noADMCI, and extended MCI (i.e., ADMCI+noADMCI) groups, several general linear models (*p* < 0.05, Bonferroni corrected) were implemented, namely one model for each MRI marker showing statistically significant differences between the noADMCI and ADMCI groups in the above statistical analysis and with CSF markers as predictors (i.e., CSF Aβ42, t-tau, and p-tau).

Finally, the third control statistical analysis was conducted using the freeware tool Jamovi Version 2.3 to test whether rsEEG source activities may be associated with the structural MRI markers of neurodegeneration in ADMCI and noADMCI patients. For the ADMCI, noADMCI, and extended MCI (i.e., ADMCI+noADMCI) groups, several general linear models (*p* < 0.05, Bonferroni corrected) were implemented, namely one model for each rsEEG marker showing statistically significant differences between the ADMCI and noADMCI groups as dependent variables, and with the MRI markers showing statistically significant differences between the ADMCI and noADMCI groups as predictors.

## 5. Conclusions

This exploratory study investigated (1) whether posterior rsEEG rhythm abnormalities may be more pronounced in ADMCI patients than in noADMCI patients with the same level of amnesic deficits and (2) the potential link between these rhythms and AD-related neuropathology and neurodegeneration in ADMCI patients. The results suggest that neurophysiological brain neural oscillatory synchronization mechanisms underpinning the generation of posterior rsEEG alpha rhythms may be more abnormal in ADMCI patients than in noADMCI patients. This abnormality may be related to brain tauopathy and parietal cortical neurodegeneration. Furthermore, it may explain, at least in part, dysfunctions in the regulation of cortical neural arousal and vigilance/conscious level in quiet wakefulness.

The study results motivate future cross-validation, with prospective, multicentric rsEEG studies in ADMCI patients with a higher number of electrodes aimed to provide further evidence in favor of the inclusion of rsEEG (pathophysiological—“P”) biomarkers in the revised A-T-N Framework of biomarkers for the assessment of AD stages [1]. These biomarkers may enrich the Precision Medicine of AD, as it is well-known that many AD patients claim disturbances in vigilance/conscious level regulation (e.g., subjective cognitive impairment, “mental fog,” and sleepiness during daytime), often without receiving any specific treatment [29]. In this application, they may also be used in association with blood biomarkers of AD neuropathology in participants at the preclinical stage of subjective cognitive deficits, carefully taking into account the possible gender effects [86].

## Figures and Tables

**Figure 1 ijms-26-00356-f001:**
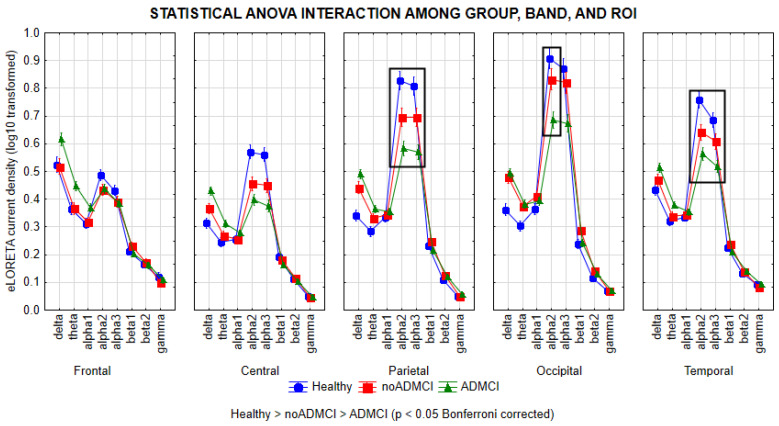
Regional normalized exact low-resolution brain electromagnetic source tomography (eLORETA) solutions (mean across subjects, log−10 transformed) modeling cortical sources of resting-state electroencephalographic (rsEEG) rhythms relative to a statistical ANOVA interaction effect (F = 6.0; *p* < 0.0001) among the factors Group (Healthy, noADMCI, and ADMCI), Band (delta, theta, alpha 1, alpha 2, alpha 3, beta 1, beta 2, and gamma), and Region of Interest, ROI (frontal, central, parietal, occipital, and temporal). This ANOVA design used the mentioned eLORETA solutions as a dependent variable. Legend: Healthy = normal older seniors; noADMCI = patients with mild cognitive impairment not due to Alzheimer’s disease; ADMCI = patients with mild cognitive impairment due to Alzheimer’s disease; the black boxes indicate the cortical regions and frequency bands in which the eLORETA solutions statistically presented a significant difference between the three groups (*p* < 0.05 corrected = *p* < 0.0004).

**Figure 2 ijms-26-00356-f002:**
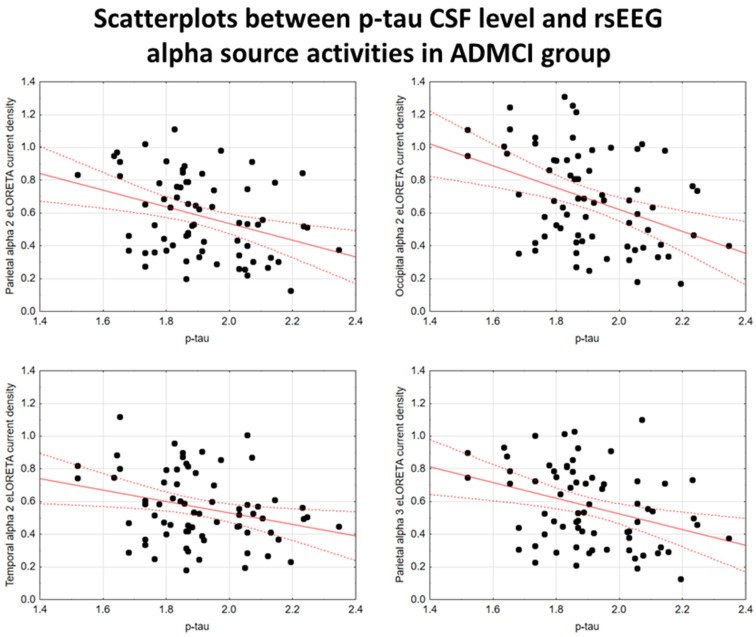
Scatterplots showing the associations between CSF p-tau levels and rsEEG alpha source activities (i.e., parietal alpha 2, occipital alpha 2, temporal alpha 2, and parietal alpha 3) in the ADMCI patients (each circle corresponds to a patient). General linear models (GLMs) evaluated the hypothesis of an association between the CSF and rsEEG variables (*p* < 0.05). Legend: CSF p-tau = phosphorylated form of tau in cerebrospinal fluid; rsEEG = resting-state electroencephalographic; ADMCI = patients with mild cognitive impairment due to Alzheimer’s disease. Dots represent individual data points, where each dot shows the values of two variables. The solid red line (line of best fit) is a line drawn through the dots to show the overall trend or relationship between the variables, indicating whether the correlation is positive, negative, or absent. The dotted red lines (Confidence lines) indicate the range where we expect the true values to fall with the 95% level of confidence.

**Figure 3 ijms-26-00356-f003:**
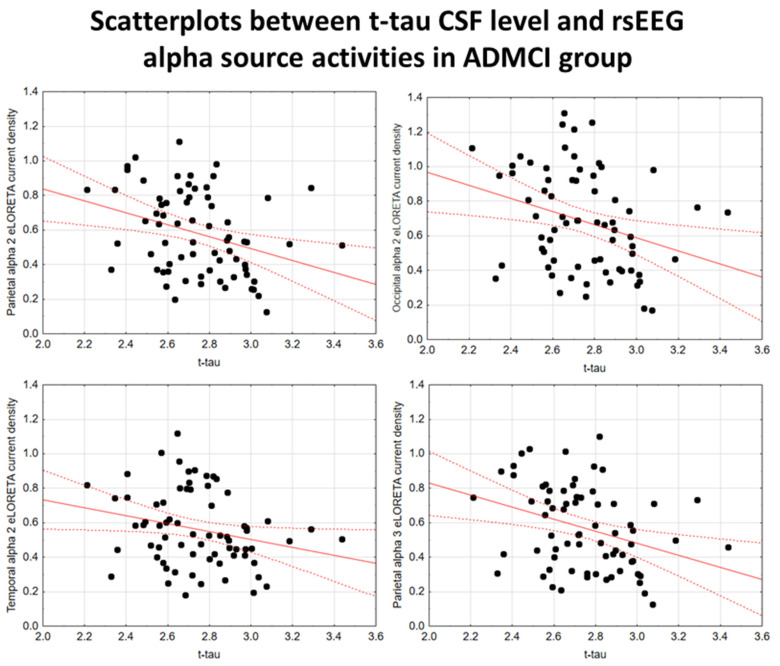
Scatterplots showing the associations between CSF t-tau levels and rsEEG alpha source activities (i.e., parietal alpha 2, occipital alpha 2, temporal alpha 2, and parietal alpha 3) in the ADMCI patients (each circle corresponds to a patient). General linear models (GLMs) evaluated the hypothesis of an association between these variables (*p* < 0.05). Legend: CSF t-tau = tau in cerebrospinal fluid; rsEEG = resting-state electroencephalographic; ADMCI = patients with mild cognitive impairment due to Alzheimer’s disease. Dots represent individual data points, where each dot shows the values of two variables. The solid red line (line of best fit) is a line drawn through the dots to show the overall trend or relationship between the variables, indicating whether the correlation is positive, negative, or absent. The dotted red lines (Confidence lines) indicate the range where we expect the true values to fall with the 95% level of confidence.

**Figure 4 ijms-26-00356-f004:**
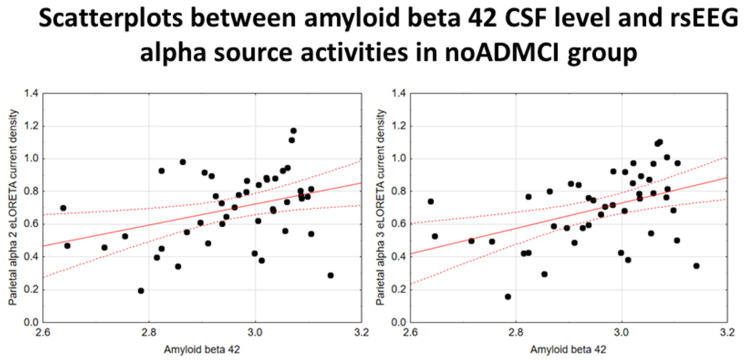
Scatterplots showing the associations between CSF Aβ42 levels and rsEEG alpha source activities (i.e., parietal alpha 2 and parietal alpha 3) in the noADMCI patients (each circle corresponds to a patient). General linear models (GLMs) evaluated the hypothesis of an association between these variables (*p* < 0.05). Legend: CSF Aβ42 = beta-amyloid 1–42 in cerebrospinal fluid; rsEEG = resting-state electroencephalographic; noADMCI = patients with mild cognitive impairment not due to Alzheimer’s disease. Dots represent individual data points, where each dot shows the values of two variables. The solid red line (line of best fit) is a line drawn through the dots to show the overall trend or relationship between the variables, indicating whether the correlation is positive, negative, or absent. The dotted red lines (Confidence lines) indicate the range where we expect the true values to fall with the 95% level of confidence.

**Figure 5 ijms-26-00356-f005:**
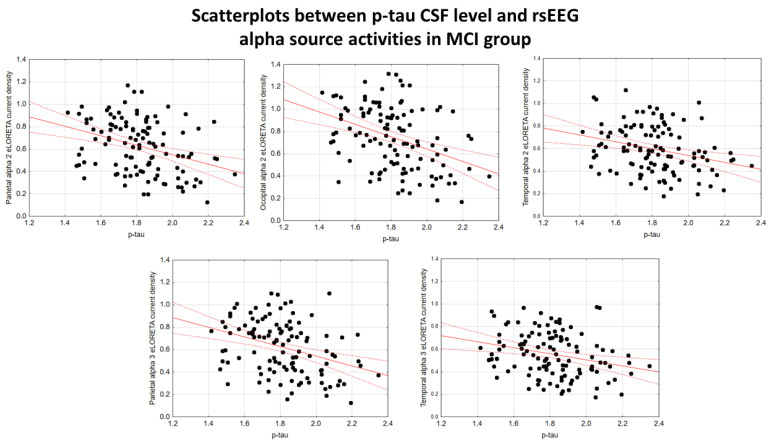
Scatterplots showing the associations between CSF p-tau levels and rsEEG alpha source activities (i.e., parietal alpha 2, occipital alpha 2, temporal alpha 2, parietal alpha 3, and temporal alpha 3) in the extended MCI (i.e., ADMCI+noADMCI) group (each circle corresponds to a patient). General linear models (GLMs) evaluated the hypothesis of an association between these variables (*p* < 0.05). Legend: CSF p-tau = phosphorylated form of tau in cerebrospinal fluid; rsEEG = resting-state electroencephalographic; ADMCI = patients with mild cognitive impairment due to Alzheimer’s disease; noADMCI = patients with mild cognitive impairment not due to Alzheimer’s disease. Dots represent individual data points, where each dot shows the values of two variables. The solid red line (line of best fit) is a line drawn through the dots to show the overall trend or relationship between the variables, indicating whether the correlation is positive, negative, or absent. The dotted red lines (Confidence lines) indicate the range where we expect the true values to fall with the 95% level of confidence.

**Figure 6 ijms-26-00356-f006:**
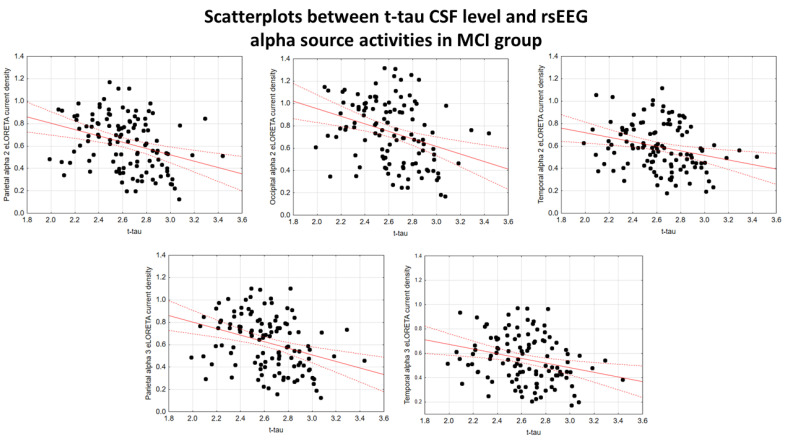
Scatterplots showing the associations between the CSF t-tau levels and the rsEEG alpha source activities (i.e., parietal alpha 2, occipital alpha 2, temporal alpha 2, parietal alpha 3, and temporal alpha 3) in the extended MCI (i.e., ADMCI+noADMCI) group (each circle corresponds to a patient). General linear models (GLMs) evaluated the hypothesis of an association between these variables (*p* < 0.05). Legend: CSF t-tau = tau in cerebrospinal fluid; rsEEG = resting-state electroencephalographic; ADMCI = patients with mild cognitive impairment due to Alzheimer’s disease; noADMCI = patients with mild cognitive impairment not due to Alzheimer’s disease. Dots represent individual data points, where each dot shows the values of two variables. The solid red line (line of best fit) is a line drawn through the dots to show the overall trend or relationship between the variables, indicating whether the correlation is positive, negative, or absent. The dotted red lines (Confidence lines) indicate the range where we expect the true values to fall with the 95% level of confidence.

**Figure 7 ijms-26-00356-f007:**
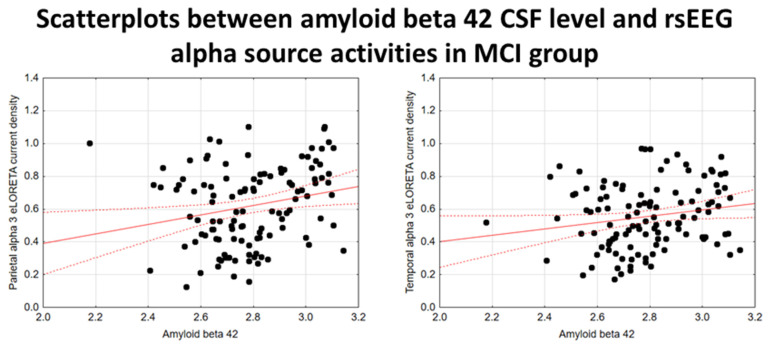
Scatterplots showing the associations between CSF Aβ42 level and rsEEG alpha source activities (i.e., parietal alpha 3 and temporal alpha 3) in the extended MCI (i.e., ADMCI+noADMCI) group (each circle corresponds to a patient). General linear models (GLMs) evaluated the hypothesis of an association between these variables (*p* < 0.05). Legend: CSF Aβ42 = beta-amyloid 1–42 in cerebrospinal fluid; rsEEG = resting-state electroencephalographic; ADMCI = patients with mild cognitive impairment due to Alzheimer’s disease; noADMCI = patients with mild cognitive impairment not due to Alzheimer’s disease. Dots represent individual data points, where each dot shows the values of two variables. The solid red line (line of best fit) is a line drawn through the dots to show the overall trend or relationship between the variables, indicating whether the correlation is positive, negative, or absent. The dotted red lines (Confidence lines) indicate the range where we expect the true values to fall with the 95% level of confidence.

**Table 1 ijms-26-00356-t001:** Mean values (±SE) of the clinical (i.e., Geriatric Depression Scale, Clinical Dementia Rating, and Hachinski Ischemic Score), genetic (i.e., Apolipoprotein E genotyping, APOE), and amyloid–tau cerebrospinal fluid (i.e., beta-amyloid 1–42, Aβ 42; protein tau, t-tau; a phosphorylated form of protein tau, p-tau) data as the results of their statistical comparisons (*p* < 0.05) in the noADMCI (N = 45) and ADMCI (N = 70) groups. The statistically significant results are reported in bold. In line with the inclusion criteria, all noADMCI and ADMCI patients had a CDR score of 0.5, a GDS score of ≤ 5, and an HIS score of ≤4. Legend: noADMCI = patients with mild cognitive impairment not due to Alzheimer’s disease; ADMCI = patients with mild cognitive impairment due to Alzheimer’s disease; n.s. = not significant (*p* > 0.05); SE = standard error of the mean.

Clinical, Genetic (APOE), and Cerebrospinal Fluid (CSF) Amyloid–Tau Markers in noADMCI and ADMCI Groups			
	noADMCI	ADMCI	Statistical Analyses
** *Clinical markers* **			
Geriatric depression scale (GDS)	2.5 ± 0.3	2.7 ± 0.3	T-test: *p* = n.s.
Clinical dementia rating (CDR)	0.5 ± 0.0	0.5 ± 0.0	T-test: *p* = n.s.
Hachinski ischemic score (HIS)	1.0 ± 0.1	0.8 ± 0.1	T-test: *p* = n.s.
** *Genetic marker* **			
APOE4 (%)	4.4%	75.7%	Fisher test: ***p* < 0.00001**
** *Cerebrospinal fluid markers* **			
Aβ42 (pg/mL)	940 ± 36	498 ± 16	T-test: ***p* < 0.00001**
p-tau (pg/mL)	49 ± 2	86 ± 4	T-test: ***p* < 0.00001**
t-tau (pg/mL)	312 ± 24	642 ± 48	T-test: ***p* < 0.00001**

**Table 2 ijms-26-00356-t002:** Mean values (±SE) of the neuropsychological scores (i.e., Logical Memory Test immediate recall, Logical Memory Test delayed recall, Rey Auditory Verbal Learning Test immediate recall, Rey Auditory Verbal Learning Test delayed recall, Trail Making Test Part B-A, Verbal Fluency for letters, Verbal Fluency for category, clock drawing, and clock copy) as well as the results of their statistical comparisons (T-test on log−10 transformed data; *p* < 0.05 corrected) in the noADMCI (N = 45) and ADMCI (N = 70) groups. Values at *p* < 0.05 are reported in bold. The cut-off scores of the neuropsychological tests are also reported. In line with the inclusion criteria, all noADMCI and ADMCI patients had Logical Memory Test scores of 1.5 standard deviations (SDs) below the mean adjusted for age. Legend: noADMCI = patients with mild cognitive impairment not due to Alzheimer’s disease; ADMCI = patients with mild cognitive impairment due to Alzheimer’s disease; RAVLT  =  Rey Auditory Verbal Learning Test; n.s. = not significant (*p* > 0.05); SE = standard error of the mean.

Neuropsychological Scores in the noADMCI and ADMCI Groups				
		noADMCI	ADMCI	
	*Cut-Off of Abnormality*	Mean ± SE *(% Subjects With Abnormal Score*)	Mean ± SE *(% Subjects with Abnormal Score*)	T-Test *p*-Value
**Logical Memory Test** **immediate recall**	<13	10.1 ± 1.4 71.1%	7.2 ± 0.5 91.0%	***p* = 0.03**
**Logical Memory Test** **delayed recall**	<12	6.2 ± 1.3 88.8%	4.6 ± 0.4 95.5%	***p* = 0.04**
**RAVLT** **immediate recall**	<28.53	32.3 ± 1.6 44.4%	29.0 ± 1.2 50%	***p* = 0.02**
**RAVLT** **delayed recall**	<4.69	4.7 ± 1.3 51.1%	3.7 ± 0.4 68.6%	***p* = 0.05**
**Trail Making Test B-A**	≥187	126.5 ± 8.2 28.4%	137.2 ± 12.1 27.0%	*p* = n.s.
**Letter fluency**	<17	27.5 ± 2.1 21.4%	32.2 ± 1.4 8.6%	*p* = n.s.
**Letter category**	<25	30.8 ± 2.1 31.8%	31.7 ± 1.2 30.9%	*p* = n.s.
**Clock drawing**	>3	3.6 ±1.3 57.1%	3.8 ± 0.2 67.1%	*p* = n.s.
**Clock copy**	>3	4.6 ± 0.1 87.1%	4.5 ± 0.1 93.0%	*p* = n.s.

**Table 3 ijms-26-00356-t003:** Results of general linear models (GLMs) assessing the association between CSF amyloid–tau markers (i.e., CSF Aβ42, t-tau, and p-tau levels; predictors) and rsEEG source activities in the noADMCI, ADMCI, and extended MCI (ADMCI+noADMCI) groups. The rsEEG markers were those showing statistically significant differences between the Healthy, noADMCI, and ADMCI groups (i.e., parietal alpha 2, occipital alpha 2, temporal alpha 2, parietal alpha 3, and temporal alpha 3 source activities estimated by eLORETA current densities from rsEEG rhythms; dependent variables). Standardized beta, t, and *p* values are reported for each GLM. Values at *p* < 0.05 are reported in bold. Legend: noADMCI = patients with mild cognitive impairment not due to Alzheimer’s disease; ADMCI = patients with mild cognitive impairment due to Alzheimer’s disease; CSF = cerebrospinal fluid; Aβ42 = beta-amyloid 1–42; t-tau= protein tau; p-tau = phosphorylated form of protein tau; rsEEG = resting-state electroencephalographic.

Group	Predictor: CSF Marker	Dependent Variable: rsEEG Source Marker	Standardized β	t	*p*
ADMCI	p-tau	Parietal alpha 2	−0.367	−3.25	**0.002**
		Occipital alpha 2	−0.399	−3.58	**0.001**
		Temporal alpha 2	−0.284	−2.45	**0.01**
		Parietal alpha 3	−0.345	−3.03	**0.003**
		Temporal alpha 3	−0.197	−1.65	0.1
	t-tau	Parietal alpha 2	−0.329	−2.87	**0.005**
		Occipital alpha 2	−0.298	−2.58	**0.01**
		Temporal alpha 2	−0.246	−2.09	**0.04**
		Parietal alpha 3	−0.332	−2.90	**0.005**
		Temporal alpha 3	−0.201	−1.69	0.1
	Aβ42	Parietal alpha 2	−0.207	−1.75	0.1
		Occipital alpha 2	−0.164	−1.37	0.2
		Temporal alpha 2	−0.081	−0.65	0.5
		Parietal alpha 3	−0.157	−1.31	0.2
		Temporal alpha 3	−0.048	−0.40	0.7
noADMCI	p-tau	Parietal alpha 2	−0.069	−0.46	0.6
		Occipital alpha 2	−0.084	−0.55	0.6
		Temporal alpha 2	−0.092	−0.60	0.5
		Parietal alpha 3	−0.069	−0.45	0.6
		Temporal alpha 3	−0.119	−0.78	0.4
	t-tau	Parietal alpha 2	−0.083	−0.54	0.5
		Occipital alpha 2	−0.107	−0.70	0.5
		Temporal alpha 2	−0.118	−0.77	0.4
		Parietal alpha 3	−0.071	−0.47	0.6
		Temporal alpha 3	−0.130	−0.86	0.4
	Aβ42	Parietal alpha 2	0.362	2.55	**0.01**
		Occipital alpha 2	0.219	1.47	0.1
		Temporal alpha 2	0.126	0.83	0.4
		Parietal alpha 3	0.436	3.17	**0.003**
		Temporal alpha 3	0.175	1.16	0.2
MCI (ADMCI+ noADMCI)	p-tau	Parietal alpha 2	−0.351	−3.98	**0.001**
		Occipital alpha 2	−0.387	−4.46	**0.001**
		Temporal alpha 2	−0.286	−3.17	**0.002**
		Parietal alpha 3	−0.351	−3.99	**0.001**
		Temporal alpha 3	−0.267	−2.94	**0.004**
	t-tau	Parietal alpha 2	−0.323	−3.62	**0.001**
		Occipital alpha 2	−0.327	−3.67	**0.001**
		Temporal alpha 2	−0.265	−2.92	**0.004**
		Parietal alpha 3	−0.333	−3.75	**0.001**
		Temporal alpha 3	−0.266	−2.94	**0.004**
	Aβ42	Parietal alpha 2	0.167	1.80	0.1
		Occipital alpha 2	0.164	1.77	0.1
		Temporal alpha 2	0.129	1.38	0.2
		Parietal alpha 3	0.224	2.45	**0.01**
		Temporal alpha 3	0.183	1.98	**0.05**

**Table 4 ijms-26-00356-t004:** Results of general linear models (GLMs) assessing the association between CSF tau markers (i.e., CSF t-tau and p-tau; predictors) and rsEEG source activities in the ADMC group. The white matter (WM) hypo-intensity from T1-weighted magnetic resonance imaging was used as a covariate reflecting subcortical WM cerebrovascular lesions. The rsEEG markers were those showing statistically significant differences between the Healthy, noADMCI, and ADMCI groups (i.e., parietal alpha 2, occipital alpha 2, temporal alpha 2, parietal alpha 3, and temporal alpha 3 source activities estimated by eLORETA current densities from rsEEG rhythms; dependent variables). Standardized beta, t, and *p* values are reported for each GLM. Values at *p* < 0.05 are reported in bold. Legend: ADMCI = patients with mild cognitive impairment due to Alzheimer’s disease; CSF = cerebrospinal fluid; t-tau= protein tau; p-tau = phosphorylated form of protein tau; rsEEG = resting-state electroencephalographic.

Group	Predictor: CSF Marker	Dependent Variable: rsEEG Source Marker	Standardized β	t	*p*
ADMCI	p-tau	Parietal alpha 2	−0.334	−2.94	**0.004**
		Occipital alpha 2	−0.358	−3.22	**0.002**
		Temporal alpha 2	−0.254	−2.65	**0.03**
		Parietal alpha 3	−0.323	−2.78	**0.007**
		Temporal alpha 3	−0.173	−1.43	0.2
	t-tau	Parietal alpha 2	−0.314	−2.78	**0.007**
		Occipital alpha 2	−0.279	−2.48	**0.01**
		Temporal alpha 2	−0.235	−2.00	**0.05**
		Parietal alpha 3	−0.325	−2.81	**0.006**
		Temporal alpha 3	−0.173	−1.43	0.2
	Aβ42	Parietal alpha 2	−0.176	−1.49	0.1
		Occipital alpha 2	−0.123	−1.05	0.3
		Temporal alpha 2	−0.049	−0.40	0.7
		Parietal alpha 3	−0.132	−1.1	0.3
		Temporal alpha 3	−0.121	−0.50	0.6

**Table 5 ijms-26-00356-t005:** Mean values (±SE) of structural MRI markers (i.e., normalized total gray matter volume, normalized total white matter volume, normalized hippocampus volume, normalized amygdala volume, mean cortical thickness, parietal cortical thickness, temporal cortical thickness, precuneus cortical thickness, and cuneus cortical thickness) in the noADMCI (N = 45) and ADMCI (N = 70) groups. The results of the statistical comparisons of those markers between the two groups are reported (T-test on log−10 transformed data; *p*  <  0.05 corrected). The statistically significant results are reported in bold. The volumes were normalized with reference to the total intracranial volume. Legend: MRI = magnetic resonance imaging; noADMCI = patients with mild cognitive impairment not due to Alzheimer’s disease; ADMCI = patients with mild cognitive impairment due to Alzheimer’s disease; n.s. = not significant (*p* > 0.05); SE = standard error of the mean.

MRI Markers in noADMCI and ADMCI Groups			
	noADMCI	ADMCI	T-Test
**Normalized global gray matter volume**	0.292 ± 0.006 SE	0.291 ± 0.003 SE	*p* = n.s.
**Normalized global white matter volume**	0.382 ± 0.006 SE	0.385 ± 0.004 SE	*p* = n.s.
**Normalized hippocampus volume**	0.0049 ± 0.0002 SE	0.0045 ± 0.0001 SE	*p* = 0.02
**Normalized amygdala volume**	0.0019 ± 0.0001 SE	0.0018 ± 0.0001 SE	*p* = n.s.
**Mean cortical thickness**	4.67 ± 0.04 SE	4.54 ± 0.03 SE	*p* = 0.02
**Parietal cortical thickness**	8.73 ± 0.10 SE	8.27 ± 0.07 SE	***p* = 0.0002**
**Temporal cortical thickness**	10.77 ± 0.12 SE	10.34 ± 0.10 SE	*p* = 0.01
**Precuneus cortical thickness**	4.43 ± 0.05 SE	4.20 ± 0.04 SE	***p* = 0.0008**
**Cuneus cortical thickness**	3.52 ± 0.04 SE	3.44 ± 0.03 SE	*p* = n.s.

**Table 6 ijms-26-00356-t006:** Results of general linear models (GLMs) assessing in the noADMCI, ADMCI, and MCI (i.e., ADMCI+noADMCI) groups the association between the CSF amyloid–tau markers (i.e., CSF Aβ42, t-tau, and p-tau levels; predictors) and the structural MRI markers that differed between the noADMCI and ADMCI groups (i.e., parietal cortical thickness and precuneus cortical thickness; dependent variables). Standardized beta, t, and *p* values are reported for each GLM. Values at *p* < 0.05 are reported in bold. Legend: noADMCI = patients with mild cognitive impairment not due to Alzheimer’s disease; ADMCI = patients with mild cognitive impairment due to Alzheimer’s disease; thk = cortical thickness; CSF = cerebrospinal fluid; Aβ42 = beta-amyloid 1–42; t-tau= protein tau; p-tau = phosphorylated form of protein tau; MRI = magnetic resonance imaging.

Group	Predictor: CSF Marker	Dependent Variable: Structural MRI Marker	Β Standardized	t	*p*
ADMCI	p-tau	Parietal cortical thk	0.031	0.25	0.8
		Precuneus cortical thk	0.021	0.17	0.8
	t-tau	Parietal cortical thk	−0.098	−0.78	0.4
		Precuneus cortical thk	−0.162	−1.31	0.2
	Aβ	Parietal cortical thk	0.263	2.17	**0.03**
		Precuneus cortical thk	0.232	1.89	0.06
noADMCI	p-tau	Parietal cortical thk	−0.194	−1.22	0.2
		Precuneus cortical thk	−0.216	−1.36	0.2
	t-tau	Parietal cortical thk	−0.146	−0.92	0.4
		Precuneus cortical thk	−0.177	−1.11	0.3
	Aβ	Parietal cortical thk	0.285	1.83	0.07
		Precuneus cortical thk	0.310	2.01	0.06
MCI (ADMCI+ noADMCI)	p-tau	Parietal cortical thk	−0.241	−2.52	**0.01**
		Precuneus cortical thk	−0.233	−2.43	**0.01**
	t-tau	Parietal cortical thk	−0.293	−3.11	**0.002**
		Precuneus cortical thk	−0.316	−3.38	**0.001**
	Aβ	Parietal cortical thk	0.428	4.81	**0.001**
		Precuneus cortical thk	0.402	4.45	**0.001**

**Table 7 ijms-26-00356-t007:** Results of general linear models (GLMs) assessing in the noADMCI, ADMCI, and MCI (ADMCI+noADMCI) groups the association between the structural MRI markers that differed between the noADMCI and ADMCI groups (i.e., parietal and precuneus cortical thicknesses; predictors) and rsEEG source activities that differed among the Healthy, noADMCI, and ADMCI groups (i.e., parietal alpha 2, occipital alpha 2, temporal alpha 2, parietal alpha 3, and temporal alpha 3 source activities estimated from rsEEG rhythms by eLORETA current densities; dependent variables). Standardized beta, t, and *p* values are reported for each GLM. Values at *p* < 0.05 are reported in bold. Legend: noADMCI = patients with mild cognitive impairment not due to Alzheimer’s disease; ADMCI = patients with mild cognitive impairment due to Alzheimer’s disease; CSF = cerebrospinal fluid; Aβ42 = beta-amyloid 1–42; t-tau= protein tau; p-tau = phosphorylated form of protein tau; rsEEG = resting-state electroencephalographic; MRI = magnetic resonance imaging.

Group	Predictor: MRI Marker	Dependent Variable: rsEEG Marker	Β Standardized	t	*p*
MCI	Parietal cortical thk	Parietal alpha 2	0.265	0.81	0.4
		Occipital alpha 2	−0.122	−0.9	0.3
		Temporal alpha 2	0.044	0.35	0.7
		Parietal alpha 3	0.153 3	1.23	0.2
		Temporal alpha 3	0.050	0.40	0.7
	Precuneus cortical thk	Parietal alpha 2	0.115	0.93	0.3
		Occipital alpha 2	−0.088	−0.71	0.5
		Temporal alpha 2	0.050	0.40	0.7
		Parietal alpha 3	0.152	1.22	0.2
		Temporal alpha 3	0.063	0.50	0.6
noADMCI	Parietal cortical thk	Parietal alpha 2	0.263	1.68	0.1
		Occipital alpha 2	0.255	1.63	0.1
		Temporal alpha 2	0.317	2.10	**0.04**
		Parietal alpha 3	0.335	2.19	**0.03**
		Temporal alpha 3	0.326	2.13	**0.04**
	Precuneus cortical thk	Parietal alpha 2	0.287	1.85	0.07
		Occipital alpha 2	0.272	1.74	0.1
		Temporal alpha 2	0.345	2.27	**0.03**
		Parietal alpha 3	0.326	2.13	**0.04**
		Temporal alpha 3	0.333	2.18	**0.03**
MCI (ADMCI+ noADMCI)	Parietal cortical thk	Parietal alpha 2	0.213	2.21	**0.02**
		Occipital alpha 2	0.081	0.83	0.4
		Temporal alpha 2	0.181	1.86	0.1
		Parietal alpha 3	0.275	2.90	**0.005**
		Temporal alpha 3	0.197	2.10	**0.04**
	Precuneus cortical thk	Parietal alpha 2	0.223	2.32	**0.02**
		Occipital alpha 2	0.098	1.00	0.3
		Temporal alpha 2	0.188	1.95	0.06
		Parietal alpha 3	0.265	2.70	**0.006**
		Temporal alpha 3	0.201	2.10	**0.04**

**Table 8 ijms-26-00356-t008:** Mean values (±SE) of the demographic and clinical data, as well as the results of their statistical comparisons (*p* < 0.05), in the groups of cognitively normal older adults (Healthy, N = 45) and patients with mild cognitive impairment not due and due to Alzheimer’s disease (noADMCI, N = 45; ADMCI, N = 70). Legend: M/F = males/females; MMSE = Mini-Mental State Evaluation; n.s. = not significant (*p* > 0.05); SE = standard error of the mean.

Demographic and Clinical Data in Healthy, noADMCI, and ADMCI Groups				
	Healthy	noADMCI	ADMCI	Statistical Analysis
**N**	45	45	70	-
**Age** **(mean years ± SE)**	68.6 ± 1.0	69.6 ± 1.2	70.0 ± 0.7	ANOVA: *p* = n.s.
**Sex** **(M/F)**	20/25	18/27	32/38	Freeman–Halton: *P* = n.s.
**Education** **(mean years ± SE)**	11.0 ± 0.6	10.0 ± 0.6	11.0 ± 0.5	ANOVA: *p* = n.s.
**MMSE** **(mean score ± SE)**	27.6 ± 0.2	25.7 ± 0.4	25.2 ± 0.2	ANOVA Kruskal–Wallis test: H = 39.1, *p* < 0.0001

## Data Availability

The data presented in this study are available upon request from the corresponding author, as some participants did not provide consent for the public sharing of their confidential data.

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
