# Peer review of "Resting-State EEG Alpha Rhythms Are Related to CSF Tau Biomarkers in Prodromal Alzheimer’s Disease"

_ijms, 2025, doi:10.3390/ijms26010356_

Round 1
Reviewer 1 Report
Comments and Suggestions for Authors
Relationship between Posterior Resting-State Electroencephalographic Alpha Rhythms and Cerebrospinal Fluid Biomarkers of Neuropathology in Patients with Alzheimer’s Disease Mild Cognitive Impairment
The study is an interesting exploratory retrospective resting state EEG investigation on AD and MCI. In the introduction, I suggest removing or make a brief reference to the consortium, and I suggest focusing more on the results. The section is well-written and clear and the hypotheses are stated in a good way at the end of the introduction.
The method is well explained and clear. However, I have some suggestions and commentaries:
The authors used GDS to assess depression/depressive symptoms in the patients. It should be interesting to see if patients have experienced MDD or mood disorders during adulthood. Moreover, the MRI was collected using different fields (1.5 and 3T), and they are from Siemens, GE, and Philips. More info about the scanners is needed, as well as the number of patients/controls collected at 1.5T and 3T. I suppose that no differences have been found, but a check is needed. Please, add this information.
Moreover, it is not clear if a 32 or 64 channels EEG system was used. You reported that 19 electrodes were considered for the analysis, but you need to specify the system that you used for resting state EEG. However, I suggest to add more information about the lasting of the EEG data collection, since you mentioned 150 epochs lasting 2 secs each.
I don’t know if you took into consideration to study EEG coherence, but it is only a suggestion and not a request to perform further analysis.
I agree with the statistical analysis and the tests that you have applied. The results are complex and I supposed that the authors selected the best option to describe them.
In the discussion, I agree that there is little info about aging and EEG (overall rsEEG), and about the speculative tone, but I suggest discussing better the role played by parietal lobe in healthy and pathological aging.
Reviewer 2 Report
Comments and Suggestions for Authors
To whom it may concern,
I enjoyed reading your manuscript. Still, some aspects could be improved (please see below my remarks):
1. The title is too long; it should be changed and be more catchy to the future reader.
2. The abstract should be structured (background; materials and method; results; conclusion)
3. In the introduction, you should add a new paragraph on mild cognitive impairment - definitions, types, current challenges. Also, the paragraphs on rEEG could be rephrased and shorten.
4. In the materials and method section, I suggest you add a new subchapter on "ethical consent".
5. In the discussion section, you could also make some speculations on the future research direction on this topic, based on your preliminary results of the study.
6. I suggest you make the conclusion section more compact, containing only two paragraphs (one summarizing the results of your work, the other suggesting future research directions).
Round 2
Reviewer 1 Report
Comments and Suggestions for Authors
The authors have addressed all my concerns and suggestions as I requested.
Reviewer 2 Report
Comments and Suggestions for Authors
The authors have addressed all of my inquiries in the correct way. The final version of the manuscript does not need any other modifications in my opinion.